# Regret Guarantees for Model-Based Reinforcement Learning with Long-Term Average Constraints

**Mridul Agarwal**[1]       **Qinbo Bai**[1]       **Vaneet Aggarwal**[2,1]

[1]School of Electrical and Computer Engineering., Purdue University, West Lafayette, Indiana, USA
[2]School of Industrial Engineering., Purdue University, West Lafayette, Indiana, USA

## Abstract

We consider the problem of constrained Markov Decision Process (CMDP) where an agent interacts with an ergodic Markov Decision Process. At every interaction, the agent obtains a reward and incurs $K$ costs. The agent aims to maximize the long-term average reward while simultaneously keeping the $K$ long-term average costs lower than a certain threshold. In this paper, we propose CMDP-PSRL, a posterior sampling based algorithm using which the agent can learn optimal policies to interact with the CMDP. We show that with the assumption of slackness, characterized by $\kappa$, the optimization problem is feasible for the sampled MDPs. Further, for MDP with $S$ states, $A$ actions, and mixing time $T_M$, we prove that following CMDP-PSRL algorithm, the agent can bound the regret of not accumulating rewards from an optimal policy by $\tilde{O}(T_M S \sqrt{AT})$. Further, we show that the violations for any of the $K$ constraints is also bounded by $\tilde{O}(T_M S \sqrt{AT})$. To the best of our knowledge, this is the first work that obtains a $\tilde{O}(\sqrt{T})$ regret bounds for ergodic MDPs with long-term average constraints using a posterior sampling method.

## 1 INTRODUCTION

Consider a wireless sensor network where the devices aim to update a server with sensor values. At time $t$, the device can choose to send a packet to obtain a reward of 1 unit or to queue the packet and obtain 0 reward. However, communicating a packet results in $p_t$ power consumption. At time $t$, if the wireless channel condition, $s_t$, is weak and the device chooses to send a packet, the resulting instantaneous power consumption, $p_t$, is high. The goal is to send as many packets as possible while keep the average power consumption, $\sum_{t=1}^{T} p_t / T$, within some limit, say $C$. This environment

has state $(s_t, q_t)$ as the channel condition and queue length at time $t$. To limit the power consumption, the agent may choose to send packets when the channel condition is good or when the queue length grows beyond a certain threshold. The agent aims to learn the policies in an *online manner* which requires efficiently balancing exploration of state-space and exploitation of the estimated system dynamics [Singh et al., 2020].

Similar to the example above, many applications require to keep some costs low while simultaneously maximizing the rewards [Altman, 1999]. Owing to the importance of this problem, in this paper, we consider the problem of constrained Markov Decision Processes (CMDP). We aim to develop a reinforcement learning algorithm following which an agent can bound the constraint violations and the regret in obtaining the optimal reward to $o(T)$.

The problem setup, where the system dynamics are known, is extensively studied [Altman, 1999]. For a constrained setup, the optimal policy is possibly stochastic [Altman, 1999, Puterman, 2014]. In the domain where the agent learns the system dynamics and aims to learn good policies online, there has been work where to show asymptotic convergence to optimal policies [Gattami et al., 2021], or even provide regret guarantees when the MDP is episodic [Zheng and Ratliff, 2020, Ding et al., 2021]. Recently, [Singh et al., 2020] considered the problem of online optimization of infinite-horizon communicating Markov Decision Processes with long-term average constraints. They provide an optimism based algorithm where confidence bounds on each transition probabilities $p(s'|s, a)$ is constructed. Using this, they obtain a regret bound of $\tilde{O}\left(\sqrt{SAT} + T_M T^{2/3}\right)$[1]. Additionally, finding the optimistic policy is a computationally intensive task as the number of optimization variables become $S^2 \times A$ for MDP with $S$ states and $A$ actions.

In this paper, we consider the reinforcement learning an infinite-horizon ergodic MDP [Tarbouriech and Lazaric,

---

[1]$\tilde{O}(\cdot)$ hides the logarithmic terms

*Accepted for the 38[th] Conference on Uncertainty in Artificial Intelligence* (UAI 2022).

2019, Gattami et al., 2021] with long-term average constraints. We use $\ell_1$ deviation bounds [Jaksch et al., 2010] and use a Bellman error analysis to bound the reward regret of the MDP as $\tilde{O}(T_M S \sqrt{AT})$. Additionally, we also bound the constraint violations as $\tilde{O}(T_M S \sqrt{AT})$. We propose a posterior sampling based algorithm where we sample the transition dynamics using a Dirichlet distribution [Osband et al., 2013], which achieves this regret bound.

Unlike optimistic algorithms, the sampled MDP may not be infeasible for the constrained optimization. Hence, we consider slackness characterized by Slater's parameter Ding et al. [2020], which allows us to prove that the optimization problem is feasible even with the sampled MDPs. Posterior sampling also helps to reduces the optimization variables, to find only the optimal policy for the sampled MDP, to only $S \times A$ variables. Finally, we provide numerical examples where the algorithm converges to the calculated optimal policies. To the best of our knowledge, this is the first work to obtain $O(\sqrt{T})$ regret guarantees for the infinite horizon long-term average constraint setup with posterior sampling.

## 2 RELATED WORK

Stochastic Optimization using Markov Decision Processes has very rich roots [Howard, 1960]. There have been work in understanding convergence of the algorithm to find optimal policies for known MDPs [Bertsekas and Tsitsiklis, 1996, Altman, 1999]. Also, when the MDP is not known, there are algorithms with asymptotic guarantees for learning the optimal policies [Watkins and Dayan, 1992] which maximize an objective without any constraints. Recent algorithms even achieve finite time near-optimal regret bounds with respect to the number of interactions with the environment [Jaksch et al., 2010, Osband et al., 2013, Agrawal and Jia, 2017, Jin et al., 2018]. Jaksch et al. [2010] uses the optimism principle for minimizing regret for weakly communicating infinite horizon MDPs with bounded diameter. Osband et al. [2013] extended the analysis of Jaksch et al. [2010] to posterior sampling for episodic MDPs and bounded the Bayesian regret and further improved the regret bounds Osband and Van Roy [2017]. Agrawal and Jia [2017] uses a posterior sampling based approach and obtains a frequentist regret for the infinite horizon MDPs with bounded diameter.

In many reinforcement learning settings, the agent not only wants to maximize the rewards but also satisfy certain cost constraints [Altman, 1999]. Early works in this area were pioneered by [Altman and Schwartz, 1991]. They provided an algorithm which combined forced explorations and following policies optimized on empirical estimates to obtain an asymptotic convergence. [Borkar, 2005] studied the constrained RL problem using actor-critic and a two time-scale framework [Borkar, 1997] to obtain asymptotic performance guarantees. Very recently, [Gattami et al., 2021] analyzed the asymptotic performance for Lagrangian based algo-

rithms for infinite-horizon long-term average constraints.

Inspired by the finite-time performance analysis of reinforcement learning algorithm for unconstrained problems, there has been a significant thrust in understanding the finite-time performances of constrained MDP algorithms. [Zheng and Ratliff, 2020] considered an episodic CMDP and use an optimism based algorithm to bound the constraint violation as $\tilde{O}(\sqrt{T^{1.5}})$ with high probability. [Kalagarla et al., 2020] also considered the episodic setup to obtain PAC-style bound for an optimism based algorithm. [Ding et al., 2021] considered the setup of $H$-episode length episodic CMDPs with $d$-dimensional linear function approximation to bound the constraint violations as $\tilde{O}(d\sqrt{H^5 T})$ by mixing the optimal policy with an exploration policy. [Efroni et al., 2020] proposes a linear-programming and primal-dual policy optimization algorithm to bound the regret as $O(S\sqrt{H^3 T})$. [Qiu et al., 2020] proposed an algorithm which obtains a regret bound of $\tilde{O}(S\sqrt{AH^2 T})$ for the problem of adversarial stochastic shortest path. Compared to these works, we focus on setting with infinite horizon long-term average constraints.

After developing a better understanding of the policy gradient algorithms [Agarwal et al., 2020], there has been theoretical work in the area of model-free policy gradient algorithms for constrained MDP and safe reinforcement learning as well. [Xu et al., 2020] consider an infinite horizon discounted setup with constraints and obtain global convergence using policy gradient algorithms. [Ding et al., 2020] also considers an infinite horizon discounted setup. They use a natural policy gradient to update the primal variable and sub-gradient descent to update the dual variable.

Recently [Singh et al., 2020] considered the setup of infinite-horizon CMDPs with long-term average constraints and obtain a regret bound of $\tilde{O}(T^{2/3})$ using an optimism based algorithm and forced explorations. We consider a similar setting with ergodic CMDP and propose a posterior sampling based algorithm to bound the regret as $\tilde{O}(poly(DSA)\sqrt{T})$ using explorations assisted by the ergodicity of the MDP.

## 3 PROBLEM FORMULATION

We consider an infinite horizon discounted Markov decision process (MDP) $\mathcal{M}$, defined by the tuple $(\mathcal{S}, \mathcal{A}, P, r, c_1, \cdots, c^k)$. $\mathcal{S}$ denotes a finite set of state space with $|\mathcal{S}| = S$, and $\mathcal{A}$ denotes a finite set of actions with $|\mathcal{A}| = A$. $P : \mathcal{S} \times \mathcal{A} \rightarrow \Delta(\mathcal{S})$ denotes the probability $P(s'|s,a)$ of transitioning to state $s'$ from state $s$ after taking action $a$. $r : \mathcal{S} \times \mathcal{A} \rightarrow [0,1]$ denotes the average reward in state $s$ after taking action $a$. $c^k : \mathcal{S} \times \mathcal{A} \rightarrow [0,1]$ denotes average cost incurred by the agent for constraint $k \in [K] = \{1, 2, \cdots, K\}$ after taking action $a$ in state $s$. We use a stochastic policy $\pi : \mathcal{S} \rightarrow \Delta(\mathcal{A})$, such that given state $s$, $\pi(a|s)$ is the probability of selecting action $a$.

Note that the a policy $\pi$ induces a Markov chain over the state space of the MDP. Pertaining to the Markov chains generated by the policies for $\mathcal{M}$, we now define the mixing time of MDP.

**Definition 1** (Mixing Time). *Consider the Markov Chain induced by the policy $\pi$ on the MDP $\mathcal{M}$. Let $T_{s \to s'}^{\pi}$ be a random variable that denotes the first time step when this Markov Chain enters state $s'$ starting from state $s$. Then, the mixing time of the MDP $\mathcal{M}$ is defined as:*

$$T_M = \max_{s' \neq s} \max_{\pi} \mathbb{E}\left[T_{s \to s'}^{\pi}\right] \tag{1}$$

Similar to Singh et al. [2020], let $P_\pi^t(s)$ be the $t$ step state distribution starting from state $s$ following policy $\pi$ and $P_\pi$ be the steady-state state distribution generated by policy $\pi$.

Our first assumption on the MDP allows any policy to reach any state $s'$ starting from any state $s$, and to converge to a steady state. We formalize the result in the following assumption:

**Assumption 1.** *The MDP $\mathcal{M}$ is ergodic, or $\|P_\pi^t(s) - P_\pi\|_{TV} \leq C\rho^t$ with $P_\pi$ being the long-term steady state distribution induced by policy $\pi$, and $C > 0$ and $\rho < 1$ are problem specific constants. And, the mixing time of the MDP $\mathcal{M}$ is finite or $T_M < \infty$.*

After discussing the transition dynamics of the system, we move to the rewards and costs of the MDP $\mathcal{M}$.

**Assumption 2.** *The reward function $r(s,a)$ and the costs $c^k(s,a), k \in [K]$ are known to the agent.*

We note that in most of the problems, rewards are engineered. Hence, Assumption 2 is justified in many setups. However, the system dynamics are stochastic and typically not known.

Following a policy $\pi$, the expected long-term average cost are given by $\zeta_\pi^{P,k}$. Also, we denote the average long-term reward using $\zeta_\pi^{P,k}$. Formally, we have:

$$\zeta_\pi^{P,k} = \mathbb{E}_{s_0,a_0,s_1,a_1,\cdots}\left[\lim_{\tau \to \infty} \frac{1}{\tau}\sum_{t=0}^{\tau} c^k(s_t,a_t)\right] \tag{2}$$

$$\lambda_\pi^{P,r} = \mathbb{E}_{s_0,a_0,s_1,a_1,\cdots}\left[\lim_{\tau \to \infty} \frac{1}{\tau}\sum_{t=0}^{\tau} r(s_t,a_t)\right] \tag{3}$$

$$s_0 \sim \rho_0(s_0), \ a_t \sim \pi(a_t|s_t), \ s_{t+1} \sim P(s_{t+1}|s_t,a_t)$$

For brevity, in the rest of the paper, $\mathbb{E}_{s_t,a_t,s_{t+1};t\geq 0}[\cdot]$ will be denoted as $\mathbb{E}_{\rho,\pi,P}[\cdot]$, where $s_0 \sim \rho_0(s_0), \ a_t \sim \pi(s_t|a_t), \ s_{t+1} \sim P(s_{t+1}|s_t,a_t)$. Both, $\zeta_\pi^{P,k}$ and $\lambda_\pi^{P,r}$ satisfy the following form of Bellman equation:

$$\lambda_\pi^{P,r} + h_\pi^{P,r}(s) = \sum_a \pi(a|s)r(s,a)$$
$$+ \sum_{s'}\sum_a \pi(a|s)P(s'|s,a)h_\pi^{P,r}(s) \tag{4}$$

$$\zeta_\pi^{P,k} + h_\pi^{P,r}(s) = \sum_a \pi(a|s)c^k(s,a) \tag{5}$$
$$+ \sum_{s'}\sum_a \pi(a|s)P(s'|s,a)h_\pi^{P,k}(s) \tag{6}$$

where $h_\pi^{P,r}(s)$ is the bias for reward and $h_\pi^{P,k}$ is the bias for cost $k \in [K]$.

The objective is find a policy $\pi^*$ which is the solution of the following optimization problem.

$$\max_\pi \lambda_\pi^{P,r} \quad \text{s.t.} \tag{7}$$
$$\zeta_\pi^{P,k} \leq C_k \quad \forall\, k \in [K] \tag{8}$$

where $C_k \ \forall\, k \in [K]$ are the bounds on the average costs which the agent needs to satisfy.

After formulating the optimization problem, we now state our next assumption characterizing the slackness.

**Assumption 3.** *There exists a policy $\pi$, and one constant $\kappa \geq 2ST_M\sqrt{14A\log(AT)/\sqrt{T}} + CST_M/((1-\rho)\sqrt{T})$ such that*

$$\zeta_\pi^{P,k} \leq C_k - \kappa \tag{9}$$

The slackness assumption is mild because, in various applications some a priori knowledge about a strictly feasible policy is available. Hence, this assumption is again a standard assumption in the constrained RL literature Efroni et al. [2020], Ding et al. [2021, 2020]. $\kappa$ is referred as Slater's constant. Ding et al. [2021] assumes that the Slater's constant $\kappa$ is known.

Any online algorithm starting with no prior knowledge will require to obtain estimates of transition probabilities $P$ and obtain reward $r$ and costs $c^k, \forall\, k \in [K]$ for each state action pair. Initially, when algorithm does not have good estimates of the model, it accumulates a regret as well as violates constraints as it does not know the optimal policy. We define reward regret $R(T)$ as the difference between the cumulative reward obtained $r_t$ vs the expected rewards from running the optimal policy $\pi^*$ for $T$ steps, or

$$R(T) = T\lambda_{\pi^*}^{P,r} - \sum_{t=1}^{T} r(s_t,a_t) \tag{10}$$

Additionally, we define constraint regret $R_k(T)$ for each constraint $k \in [K]$ as the gap between the cumulative cost incurred $c_t^k, k \in [K]$ and constraint bounds, or

$$R^k(T) = \left(\sum_{t=1}^{T} c^k(s_t,a_t) - TC_k\right)_+, \tag{11}$$

where $(x)_+ = \max(0,x)$.

In the following section, we present a model-based algorithm to obtain this policy $\pi^*$, and reward regret and the constraint regret accumulated by the algorithm.

# 4 THE CMDP-PSRL ALGORITHM

For infinite horizon optimization problems (or $\tau \to \infty$), we can use steady state distribution of the state to obtain expected long-term rewards or costs [Puterman, 2014]. We use

$$\zeta_\pi^{P,k} = \sum_{s\in\mathcal{S}}\sum_{a\in\mathcal{A}} c_k(s,a)d_\pi^P(s,a), \ \forall k \in [K] \quad (12)$$

$$\lambda_\pi^{P,r} = \sum_{s\in\mathcal{S}}\sum_{a\in\mathcal{A}} r(s,a)d_\pi^P(s,a) \quad (13)$$

where $d_\pi^P(s,a)$ is the steady state joint distribution of the state and actions under policy $\pi$.

Based on the above formulation, we can solve the joint optimization problem of following form

$$\max_{d(s,a)} \sum_{s\in\mathcal{S}}\sum_{a\in\mathcal{A}} r(s,a)d(s,a) \quad (14)$$

with the following set of constraints,

$$\sum_{a\in\mathcal{A}} d(s',a) = \sum_{s\in\mathcal{S},a\in\mathcal{A}} P(s'|s,a)d(s,a) \quad (15)$$

$$\sum_{s\in\mathcal{S},a\in\mathcal{A}} d(s,a) = 1, \ d(s,a) \geq 0 \quad (16)$$

$$\sum_{s\in\mathcal{S}}\sum_{a\in\mathcal{A}} c_k(s,a)d(s,a) \leq C_k \ \forall k \in [K] \quad (17)$$

for all $s' \in \mathcal{S}$, $\forall s \in \mathcal{S}$, and $\forall a \in \mathcal{A}$. Equation (15) denotes the constraint on the transition structure for the underlying Markov Process. Equation (16) ensures that the solution is a valid probability distribution. Finally, Equation (17) are the constraints for the constrained MDP setup which the policy must satisfy.

Note that arguments in Equation (14) are linear, and the constraints in Equation (15) and Equation (16) are linear, this is a linear programming problem. Since convex optimization problems can be solved in polynomial time [Potra and Wright, 2000], we can use standard approaches to solve Equation (14). After solving the optimization problem, we obtain the optimal policy from the obtained steady state distribution $d^*(s,a)$ as,

$$\pi^*(a|s) = \frac{\mathbb{P}(s,a)}{\mathbb{P}(s)} = \frac{d^*(s,a)}{\sum_{b\in\mathcal{A}} d^*(s,b)} \ \forall s \in \mathcal{S} \quad (18)$$

Since we assumed that the CMDP is ergodic, the Markov Chain induced from policy $\pi$ is ergodic. Hence, every state is reachable following the policy $\pi^*$, we have $\mathbb{P}(s) > 0$ and Equation (18) is defined for all states $s \in \mathcal{S}$.

Further, since we assumed that the induced Markov Chain is irreducible for all stationary policies, we assume Dirichlet distribution as prior for the state transition probability $P(s'|s,a)$. Dirichlet distribution is also used as a standard

prior in literature [Agrawal and Jia, 2017, Osband et al., 2013]. Further, there exists a steady state distribution when the transition probability is sampled from a Dirichlet distribution [Agarwal et al., 2022, Proposition 1].

The complete constrained posterior sampling based algorithm, which we name CMDP-PSRL, is described in Algorithm 1. The algorithm proceeds in epochs, and a new epoch is started whenever the visitation count in epoch $e$, $\nu_e(s,a)$, is at least the total visitations before episode $e$, $N_e(s,a)$, for any state action pair (Line 8). In Line 9, we sample transition probabilities $\tilde{P}$ using the updated posterior and in Line 10, we update the policy using the optimization problem specified in Equation (14)-(17) for $P = \tilde{P}_e$. Further, if the sampled MDP does not satisfy the cost constraint in Equation (17), we ignore that constraint [2] for that epoch.

---

**Algorithm 1** CMDP-PSRL

---

1: **Input:** $\mathcal{S}, \mathcal{A}, r, c_1, \cdots, c_K$
2: Initialize $N(s,a,s') = 1 \ \forall(s,a,s') \in \mathcal{S}\times\mathcal{A}\times\mathcal{S}$, $\pi_e(a|s) = \frac{1}{|\mathcal{A}|} \ \forall \ (a,s) \in \mathcal{A} \times \mathcal{S}$, $e = 0$, $\nu_e(s,a) = N_e(s,a) = 0 \ \forall(s,a) \in \mathcal{S} \times \mathcal{A}$
3: **for** time index $t = 1, 2, \cdots$ **do**
4:     Observe state $s$
5:     Play action $a \sim \pi(\cdot|s)$
6:     Observe rewards $\{r^k\}$ and next state $s'$
7:     $\nu_e(s,a)+=1$, $N(s,a,s')+=1$
8:     **if** $\nu_e(s,a) \geq \max(1, N_e(s,a))$ for any $s,a$ **then**
9:         $\tilde{P}_e(s'|a,s) \sim Dir(N(s,a,s')) \ \forall \ (s,a,s')$
10:         Solve steady state distribution $d(s,a)$ as the solution of the optimization problem in Equations (14-17) for $\tilde{P}_e$.
11:         Obtain optimal policy for next epoch, $e+1$, $\pi_{e+1}$ as

$$\pi_{e+1}(a|s) = \frac{d(s,a)}{\sum_{a\in\mathcal{A}} d(s,a)}$$

12:         $e = e+1$
13:         $t_e = t$
14:         $\nu_e(s,a) = 0, N_e(s,a) = \sum_{e'}^e \nu_{e'}(s,a) \ \forall(s,a)$
15:     **end if**
16: **end for**

---

# 5 ANALYSIS

We first obtain the feasibility of the optimization problem Equation (14)-(17) for the sampled MDP. We note that we assumed slackness in the true MDP with transition probabilities $P$. Hence, if the the sampled MDP is close to the true MDP, the deviation in the cost will be less and there will be a policy which satisfies the constraint in Equation (17). We formalize this intuition in the following result.

**Lemma 1.** *Following Algorithm 1, if $t_{e+1} - t_e \geq \sqrt{T}$ and $\|\tilde{P}_e(\cdot|s,a) - P(\cdot|s,a)\|_1 \leq \sqrt{\frac{14S\log(2At)}{N_e(s,a)}} \forall \ s, a$ there exists*

---

[2]We will show in the analysis that cumulative constraint violations are still bounded.

*a policy $\pi$ which satisfies,*

$$\zeta_\pi^{\tilde{P}_e,k} \le C_k \ \forall \ k \in [K], \tag{19}$$

*and the optimization problem in Equation (14)-(17) is feasible, where $t_e$ is the start time of epoch $e$.*

*Proof Outline.* We consider the policy $\pi$ which satisfies the Slater's condition in Equation (9). We then consider the Bellman error of taking one step in MDP with transition probabilities $\tilde{P}_e$ and then following policy $\pi$ on the MDP with transition probabilities $P$. Now, using [Agarwal et al., 2022, Lemma 1] relating the average costs following policy $\pi$ with $P$ and $\tilde{P}_e$ ($\zeta_\pi^{P,k}$, and $\zeta_\pi^{\tilde{P}_e,k}$ for all $k \in [K]$ respectively) with the Bellman error gives the required result. The complete proof is provided in the supplementary material. $\square$

After obtaining a feasible policy $\pi_e$ maximizing rewards for the sampled MDP, we now quantify is regret. We note that when optimizing for long-term average rewards and long-term average constraints, we want to simultaneously minimize the reward regret and the constraint regrets. Further, if we know the optimal policy $\pi^*$ before hand, the deviations resulting from the stochasticity of the process can still result in some constraint violations. Also, since we sample a MDP, the policy which is feasible for the MDP may violate constraints on the true MDP. We want to bound this gap between $K$ costs for the two MDPs as well.

We aim to quantify the regret from **(R.1)** deviation of long-term average rewards of the optimal policy because of incorrect knowledge of the MDP ($\lambda_{\pi^*}^{P,r} - \lambda_{\pi_e}^{\tilde{P}_e,r}$), **(R.2)** deviation of the long-term average rewards generated by the optimal policy for the sampled MDP on the sampled MDP and the long-term average rewards generated by the optimal policy for the sampled MDP on the true MDP ($\lambda_{\pi_e}^{\tilde{P}_e,r} - \lambda_{\pi_e}^{P,r}$), and **(R.3)** deviation of the expected rewards from following the optimal policy of the sampled MDP ($\lambda_{\pi_e}^{P,r} - r(s_t, a_t)$).

Similarly, the constraint violations for each $k \in [K]$ are incurred from **(C.1)** deviation of long-term average rewards of the optimal policy because of incorrect knowledge of the MDP ($C_k - \zeta_{\pi_e}^{\tilde{P}_e,k}$), **(C.2)** deviation of the long-term average costs generated by the optimal policy for the sampled MDP on the sampled MDP and the long-term average costs generated by the optimal policy for the sampled MDP on the true MDP ($\zeta_{\pi_e}^{\tilde{P}_e,r} - \lambda_{\pi_e}^{P,r}$), and **(C.3)** deviation of the expected costs from following the optimal policy of the sampled MDP ($\zeta_{\pi_e}^{P,r} - c^k(s_t, a_t)$).

We now prove the regret bounds for Algorithm 1. We first give the high level ideas used in obtaining the bounds on regret. We divide the regret into regret incurred in each epoch $e$. Then, we use the posterior sampling lemma [Osband et al., 2013, Lemma 1] to obtain the equivalence between the long-term average rewards of the true MDP $\mathcal{M}$ and the long-term average rewards for the optimal value of the sampled MDP

$\widehat{\mathcal{M}}$. This step allows us to deal with the regret from **(R.1)**. Then we use the Bellman error formulation to relate average rewards for the policy $\pi_e$ on $P$ and $\tilde{P}_e$ [Agarwal et al., 2022]. Combining this with Azuma's concentration inequality for Martingales allows us to bound the regret from **(R.2)** and **(R.3)**.

Bounding constraint violations requires similar considerations for **(C.2)** and **(C.3)**. Further, **(C.1)** becomes zero if Equation (17) is feasible for the sampled MDP. However, if Equation (17) is not feasible, the cost may be as high as $1$ ($c^k(s,a) \le 1 \ \forall \ k \in [K]$). We bound the violations by bounding the time-steps for which the optimal policy for unconstrained optimization runs.

To obtain bounds on the regret, we first note that the total number of epochs, $E$, for which the Algorithm 1 runs is bounded by $O(1 + 2SA + SA \log(T))$ from [Jaksch et al., 2010, Proposition 1].

We formally state the regret bounds and constraint violation bounds in Theorem 1 which we prove rigorously in the supplementary material.

**Theorem 1.** *The expected reward regret $\mathbb{E}[R(T)]$, and the expected constraint regret $\mathbb{E}[R_k(T)] \ \forall \ k \in [K]$ of Algorithm 1 are bounded as*

$$\mathbb{E}[R(T)] \le O\left(T_M S \sqrt{AT \log(AT)} + \frac{CS^2 A \log T}{1 - \rho}\right)$$

$$\mathbb{E}[R^k(T)] \le O\left(T_M S \sqrt{AT \log(AT)} + \frac{CS^2 A \log T}{1 - \rho}\right)$$

*Proof Outline.* We break the cumulative regret into the regret incurred in each epoch $e$. This gives us:

$$\mathbb{E}[R_T] = \mathbb{E}\left[\sum_{e=1}^{E} \sum_{t=t_e}^{t_{e+1}-1} \left(\lambda_{\pi^*}^{P,r} - r(s_t, a_t)\right)\right] \tag{20}$$

$$= \sum_{e=1}^{E} \mathbb{E}\left[\sum_{t=t_e}^{t_{e+1}-1} \left(\lambda_{\pi^*}^{P,r} - r(s_t, a_t)\right)\right] \tag{21}$$

$$= \sum_{e=1}^{E} \mathbb{E}\left[\sum_{t=t_e}^{t_{e+1}-1} \left(\lambda_{\pi_e}^{\tilde{P}_e,r} - r(s_t, a_t)\right)\right] \tag{22}$$

$$= \sum_{e=1}^{E} \mathbb{E}\left[\sum_{t=t_e}^{t_{e+1}-1} \left(\lambda_{\pi_e}^{\tilde{P}_e,r} - \lambda_{\pi_e}^{P,r} + \lambda_{\pi_e}^{P,r} - r(s_t, a_t)\right)\right]$$

$$= \sum_{e=1}^{E} \mathbb{E}\left[\sum_{t=t_e}^{t_{e+1}-1} \left(\lambda_{\pi_e}^{\tilde{P}_e,r} - \lambda_{\pi_e}^{P,r}\right)\right]$$

$$+ \mathbb{E}\left[\sum_{e=1}^{E} \sum_{t=t_e}^{t_{e+1}-1} \left(\lambda_{\pi_e}^{P,r} - r(s_t, a_t)\right)\right] \tag{23}$$

The Equation (22) follows from [Osband et al., 2013, Lemma 1] for regret each each epoch of Equation (21). Proceeding from Equation (23) requires additional consideration. Typical proof techniques to bound regret requires a

bounded bias-span $(\max_{s,s'}(h_\pi^{\tilde{P}_e,r}(s) - h_\pi^{\tilde{P}_e,r}(s')))$ which may be large for the sampled MDP. For this, we consider an MDP for the transition probability $P_e^r$ satisfies

$$\lambda_{\pi_e}^{P_e^r,r} \geq \max_{P' \in \mathcal{P}_{t_e}} \lambda_{\pi_e}^{P',r}, \text{ where} \tag{24}$$

$$\mathcal{P}_{t_e} = \left\{ P' : \|P'(\cdot|s,a) - \bar{P}_{t_e}(\cdot|s,a)\|_1 \right.$$
$$\left. \leq \sqrt{\frac{14S \log(AT)}{N_e(s,a)}} \right\} \forall\, s, a$$

where $\bar{P}_{t_e}(\cdot|s,a)$ is the estimated transition probability given $s, a$ at time $t_e$. We now have,

$$R(T) \leq \sum_{e=1}^{E} \mathbb{E}\left[ \sum_{t=t_e}^{t_{e+1}-1} \left( \lambda_{\pi_e}^{P_e^r,r} - \lambda_{\pi_e}^{P,r} \right) \right]$$
$$+ \sum_{e=1}^{E} \mathbb{E}\left[ \sum_{t=t_e}^{t_{e+1}-1} \left( \lambda_{\pi_e}^{P,r} - r(s_t, a_t) \right) \right] \tag{25}$$

The first term of Equation (25) is bounded by bounding the expected Bellman error. The second term is converted to a Martingale sequence by conditioning it on the state $s_{t_e}$ and is bounded using the ergodicity of the MDP $\mathcal{M}$ and Azuma's concentration inequality. The complete proof on bounding the regret is provided in the supplementary material.

Regarding the constraint violations, for each $k \in [K]$, we want to bound,

$$\mathbb{E}\left[ R^k(T) \right] = \mathbb{E}\left[ \left( \sum_{t=1}^{T} c_k(s_t, a_t) - TC_k \right)_+ \right] \tag{26}$$

We divide the constraint violation regret into regret over epochs as well. Now, for each epoch, we know that the constraint is satisfied by the policy for the sampled MDP. This allows us to obtain:

$$\mathbb{E}\left[ R^k(T) \right] = \mathbb{E}\left[ \left( \sum_{e} \sum_{t=t_e}^{t_{e+1}-1} (c_k(s_t, a_t) - C_k) \right)_+ \right] \tag{27}$$

$$= \mathbb{E}\left[ \left( \sum_{e} \sum_{t=t_e}^{t_{e+1}-1} \left( (c_k(s_t, a_t) - \zeta_{\pi_e}^{P,k}) \right. \right. \right.$$
$$\left. \left. \left. + \left( \zeta_{\pi_e}^{P,k} - \zeta_{\pi_e}^{\tilde{P}_e,k} \right) + \left( \zeta_{\pi_e}^{\tilde{P}_e,k} - C_k \right) \right) \right)_+ \right] \tag{28}$$

$$= \mathbb{E}\left[ \left( \sum_{e} \sum_{t=t_e}^{t_{e+1}-1} c_k(s_t, a_t) - \zeta_{\pi_e}^{P,k} \right)_+ \right.$$
$$\left. + \left( \sum_{e} \sum_{t=t_e}^{t_{e+1}-1} \zeta_{\pi_e}^{P,k} - \zeta_{\pi_e}^{\tilde{P}_e,k} \right)_+ \right.$$

$$\left. + \left( \sum_{e} \sum_{t=t_e}^{t_{e+1}-1} \zeta_{\pi_e}^{\tilde{P}_e,k} - C_k \right)_+ \right] \tag{29}$$

$$= \mathbb{E}\left[ \left| \sum_{e} \sum_{t=t_e}^{t_{e+1}-1} \left( c_k(s_t, a_t) - \zeta_{\pi_e}^{P,k} \right) \right| \right.$$
$$+ \left| \sum_{e} \sum_{t=t_e}^{t_{e+1}-1} \zeta_{\pi_e}^{P,k} - \zeta_{\pi_e}^{\tilde{P}_e,k} \right|$$
$$\left. + \left( \sum_{e} \sum_{t=t_e}^{t_{e+1}-1} \zeta_{\pi_e}^{\tilde{P}_e,k} - C_k \right)_+ \right] \tag{30}$$

The first term in Equation (28) denotes the difference between the incurred costs and the expected costs from following policy $\pi_e$. The second term denotes the difference between the expected costs from policy $\pi_e$ on the true MDP and on the sampled MDP. The third terms denotes the violations of the policy $\pi_e$ which would be zero if the policy $\pi_e$ satisfies constraint Eqution (17) for the sampled MDP. Equation (29) is obtained from the fact $\max(0, x + y) \leq \max(0, x) + \max(0, y)$ and Equation (28) is obtained from the fact $\max(0, x) \leq |x|$.

The first and second term in Equation (28) are bounded similar to Equation (23), and we focus our attention to the third term. If the optimization problem in Equation (14)-(17) is feasible, the term $(\zeta_{\pi_e}^{\tilde{P}_e,k} - C_k) \leq 0$ and if the optimization equation is infeasible, the term is upper bounded by 1 as $C_k \geq 0$ and $\zeta_{\pi_e}^{\tilde{P}_e} \leq 1$. Hence, we get:

$$\left( \sum_{e} \sum_{t=t_e}^{t_{e+1}-1} \left( \zeta_{\pi_e}^{\tilde{P}_e,k} - C_k \right) \right)_+$$

$$\leq \sum_{e} \left( \sum_{t=t_e}^{t_{e+1}-1} \zeta_{\pi_e}^{\tilde{P}_e,k} - C_k \right)_+ \tag{31}$$

$$= \sum_{e} \left( \sum_{t=t_e}^{t_{e+1}-1} \zeta_{\pi_e}^{\tilde{P}_e,k} - C_k \right)_+ \mathbf{1}\left\{ t_{e+1} - t_e > \sqrt{T} \right\}$$
$$+ \sum_{e} \left( \sum_{t=t_e}^{t_{e+1}-1} \zeta_{\pi_e}^{\tilde{P}_e,k} - C_k \right)_+ \mathbf{1}\left\{ t_{e+1} - t_e \leq \sqrt{T} \right\} \tag{32}$$

$$\leq \sum_{e} \sum_{t=t_e}^{t_{e+1}-1} \mathbf{1}\left\{ t_{e+1} - t_e \leq \sqrt{T} \right\} \tag{33}$$

$$\leq \sum_{e} \sqrt{T} = E\sqrt{T} \tag{34}$$

$$\leq (1 + 2SA + SA \log_2(T/SA))\sqrt{T} \tag{35}$$

where Equation (31) follows from the fact that total violations are less than the cumulative violations are considered per epoch. Equation (33) follows from Lemma 1 as $\left( \zeta_{\pi_e}^{\tilde{P}_e,k} - C_k \right) \leq 0$ when $t_e > \sqrt{T}$ and Equation (35)

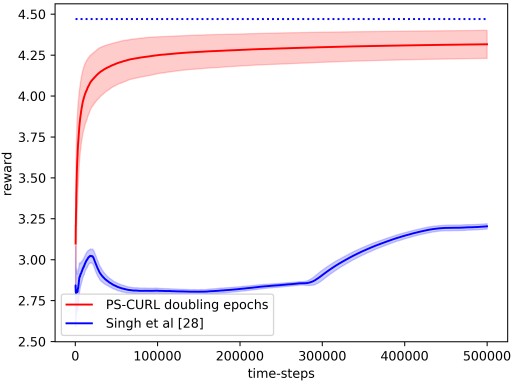

(a) Reward growth *w.r.t.* time

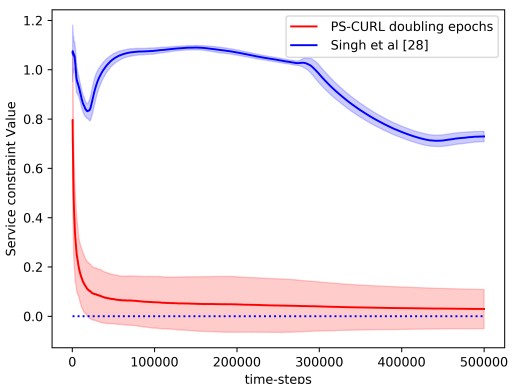

(a) Service constraints *w.r.t.* time

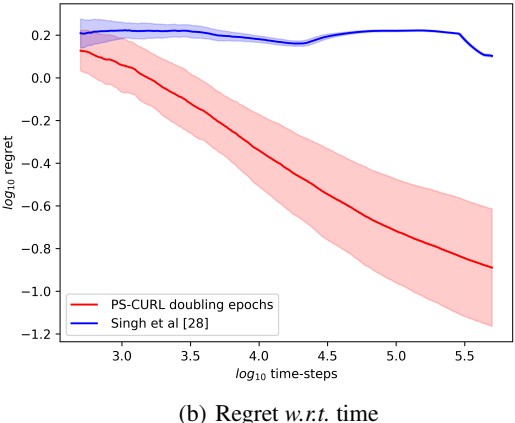

(b) Regret *w.r.t.* time

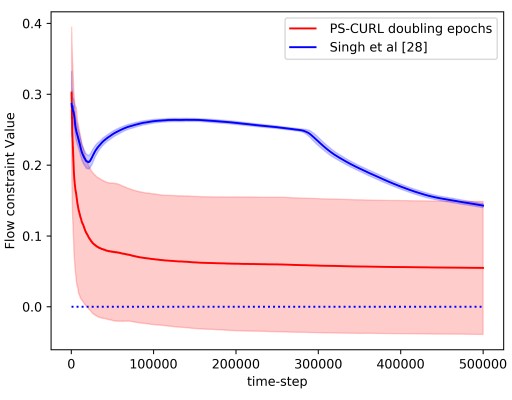

(b) Flow constraints *w.r.t.* time

Figure 1: Reward and regret performance of the proposed CMDP-PSRL algorithm on a flow and service control problem for a single queue. The algorithms is compared against the optimistic algorithm from Singh et al. Singh et al. [2020] compared to which our algorithm extremely well.

Figure 2: Constraint violation performance of the proposed CMDP-PSRL algorithm on a flow and service control problem for a single queue. The average constraint violations become zero as the algorithm proceeds, however, it never crosses zero to increase the reward further.

comes from [Jaksch et al., 2010, Proposition 1]. $\qquad\square$

We note that the fundamental setup of unconstrained optimization ($K = 0$), the bound is loose compared to that of UCRL2 algorithm Jaksch et al. [2010]. This is because we use a stochastic policy instead of a deterministic policy. Recall that the optimal policy for CMDP setup is possibly stochastic Altman [1999].

## 6 EVALUATION OF THE PROPOSED ALGORITHM

To validate the performance of the proposed CDMP-PSRL algorithm and the understanding of our analysis, we run the simulation on the flow and service control in a single-serve queue, which is introduced in [Altman and Schwartz,

1991]. A discrete-time single-server queue with a buffer of finite size $L$ is considered in this case. The number of the customer waiting in the queue is considered as the state in this problem and thus $|S| = L + 1$. Two kinds of the actions, service and flow, are considered in the problem and control the number of customers together. The action space for service is a finite subset $A$ in $[a_{min}, a_{max}]$, where $0 < a_{min} \leq a_{max} < 1$. Given a specific service action $a$, the service a customer is successfully finished with the probability $b$. If the service is successful, the length of the queue will reduce by 1. Similarly, the space for flow is also a finite subsection $B$ in $[b_{min}, b_{max}]$. In contrast to the service action, flow action will increase the queue by 1 with probability $b$ if the specific flow action $b$ is given. Also, we assume that there is no customer arriving when the queue is full. The overall action space is the Cartesian product of the $A$ and $B$. According to the service and flow probability, the

transition probability can be computed and is given in the Table 1.

For the reward function as $r(s, a, b)$ and the constraints for service and flow as $c^1(s, a, b)$ and $c^2(s, a, b)$, respectively, and stationary policies for service and flow as $\pi_a$ and $\pi_b$, respectively, the problem can be defined as

$$\max_{\pi_a, \pi_b} \quad \lim_{T \to \infty} \frac{1}{T} \sum_{t=1}^{T} r(s_t, \pi_a(s_t), \pi_b(s_t))$$

$$s.t. \quad \lim_{T \to \infty} \frac{1}{T} \sum_{t=1}^{T} c^1(s_t, \pi_a(s_t), \pi_b(s_t)) \geq 0 \quad (36)$$

$$\lim_{T \to \infty} \frac{1}{T} \sum_{t=1}^{T} c^2(s_t, \pi_a(s_t), \pi_b(s_t)) \geq 0$$

According to the discussion in [Altman and Schwartz, 1991], we define the reward function as $r(s, a, b) = 5 - s$, which is an decreasing function only dependent on the state. It is reasonable to give higher reward when the number of customer waiting in the queue is small. For the constraint function, we define $c^1(s, a, b) = -10a + 6$ and $c^2 = -8 * (1 - b)^2 + 2$, which are dependent only on service and flow action, respectively. Higher constraint value is given if the probability for the service and flow are low and high, respectively.

In the simulation, the length of the buffer is set as $L = 5$. The service action space is set as $[0.2, 0.4, 0.6, 0.8]$ and the flow action space is set as $[0.4, 0.5, 0.6, 0.7]$. We use the length of horizon $T = 50000$ and run $50$ independent simulations of the proposed CMDP-PSRL algorithm. We also plot the standard deviation around the mean value in the shadow to show the random error. In order to compare this result to the optimal, we assume that the full information of the transition dynamics is known and then use Linear Programming to solve the problem. The optimal cumulative reward from LP is shown to be $4.47$. The reward performance of the CMDP-PSRL algorithm is shown in the Figure 1 where we observe that the reward converges towards the optimal value. We also plot the constraint violations in Figure 2. The service and flow constraints converge to 0 as expected. We note that the reward of the proposed CMDP-PSRL algorithm becomes closer the optimal reward as the algorithm proceeds, and to further increase the reward, it does not violates the constraint.

We also compared our algorithm against the optimistic algorithm of Singh et al. [2020]. We note that their algorithm performs significantly worse compared to our algorithm. We account this poor performance on two accounts. An optimistic algorithm does not find a policy for transition probabilities close to $P$ for significantly large time. The other issue is because they consider confidence interval for each $P(s'|s, a)$. This also shows in their analysis and hence they obtain a $O(T^{2/3})$ regret bound. Further, the optimiza-

tion problem takes a significantly more time to solve for optimistic setup. However, the variance of their optimistic algorithm is significantly lower compared to the variance of our CMDP-PSRL algorithm.

## 7 CONCLUSION

This paper, considers the setup of reinforcement learning in ergodic infinite-horizon constrained Markov Decision Processes with $K$ long-term average constraint. We propose a posterior sampling based algorithm, CMDP-PSRL, which proceeds in epochs. At every epoch, we sample a new CMDP and generate a solution for the constraint optimization problem. A major advantage of the posterior sampling based algorithm over an optimistic approach is, that it reduces the complexity of solving for the optimal solution of the constraint problem. We also study the proposed CMDP-PSRL algorithm from regret perspective. We bound the regret of the reward collected by the CMDP-PSRL algorithm as $\tilde{O}(T_M S\sqrt{AT} + CS^2 A/(1 - \rho))$. Further, we bound the gap between the long-term average costs of the sampled MDP and the true MDP to bound the $K$ constraint violations as $\tilde{O}(T_M S\sqrt{AT} + CS^2 A/(1 - \rho))$. Finally, we evaluate the proposed CMDP-PSRL algorithm on a flow control problem for single queue and show that the proposed algorithm performs empirically well. This paper is the first work which obtains a $\tilde{O}(\sqrt{T})$ regret bounds for ergodic MDPs with long-term average constraints using a posterior sampling algorithm. A model-free algorithm that obtains similar regret bounds for infinite horizon long-term average constraints remains an open problem.

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

Table 1: Transition probability of the queue system

| Current State | $P(x_{t+1} = x_t - 1)$ | $P(x_{t+1} = x_t)$ | $P(x_{t+1} = x_t + 1)$ |
|---|---|---|---|
| $1 \leq x_t \leq L - 1$ | $a(1-b)$ | $ab + (1-a)(1-b)$ | $(1-a)b$ |
| $x_t = L$ | $a$ | $1 - a$ | $0$ |
| $x_t = 0$ | $0$ | $1 - b(1-a)$ | $b(1-a)$ |

E. Altman and A. Schwartz. Adaptive control of constrained markov chains. *IEEE Transactions on Automatic Control*, 36(4):454–462, 1991. doi: 10.1109/9.75103.

Eitan Altman. *Constrained Markov decision processes*, volume 7. CRC Press, 1999.

D. P. Bertsekas and J. N. Tsitsiklis. *Neuro-dynamic programming*. Athena Scientific, Belmont, MA, 1996.

Vivek S. Borkar. Stochastic approximation with two time scales. *Systems & Control Letters*, 29(5):291–294, 1997. ISSN 0167-6911. doi: https://doi.org/10.1016/S0167-6911(97)90015-3. URL https://www.sciencedirect.com/science/article/pii/S0167691197900153.

V.S. Borkar. An actor-critic algorithm for constrained markov decision processes. *Systems & Control Letters*, 54(3):207–213, 2005. ISSN 0167-6911. doi: https://doi.org/10.1016/j.sysconle.2004.08.007. URL https://www.sciencedirect.com/science/article/pii/S0167691104001276.

Dongsheng Ding, Kaiqing Zhang, Tamer Basar, and Mihailo Jovanovic. Natural policy gradient primal-dual method for constrained markov decision processes. *Advances in Neural Information Processing Systems*, 33, 2020.

Dongsheng Ding, Xiaohan Wei, Zhuoran Yang, Zhaoran Wang, and Mihailo Jovanovic. Provably efficient safe exploration via primal-dual policy optimization. In *International Conference on Artificial Intelligence and Statistics*, pages 3304–3312. PMLR, 2021.

Yonathan Efroni, Shie Mannor, and Matteo Pirotta. Exploration-exploitation in constrained mdps. *arXiv preprint arXiv:2003.02189*, 2020.

Ather Gattami, Qinbo Bai, and Vaneet Aggarwal. Reinforcement learning for constrained markov decision processes. In *International Conference on Artificial Intelligence and Statistics*, pages 2656–2664. PMLR, 2021.

R. A. Howard. *Dynamic Programming and Markov Processes*. MIT Press, Cambridge, MA, 1960.

Thomas Jaksch, Ronald Ortner, and Peter Auer. Near-optimal regret bounds for reinforcement learning. *Journal of Machine Learning Research*, 11(Apr):1563–1600, 2010.

Chi Jin, Zeyuan Allen-Zhu, Sebastien Bubeck, and Michael I Jordan. Is q-learning provably efficient? In S. Bengio, H. Wallach, H. Larochelle, K. Grauman, N. Cesa-Bianchi, and R. Garnett, editors, *Advances in Neural Information Processing Systems*, volume 31. Curran Associates, Inc., 2018. URL https://proceedings.neurips.cc/paper/2018/file/d3b1fb02964aa64e257f9f26a31f72cf-Paper.pdf.

Krishna C Kalagarla, Rahul Jain, and Pierluigi Nuzzo. A sample-efficient algorithm for episodic finite-horizon mdp with constraints. *arXiv preprint arXiv:2009.11348*, 2020.

Ian Osband and Benjamin Van Roy. Why is posterior sampling better than optimism for reinforcement learning? In *International conference on machine learning*, pages 2701–2710. PMLR, 2017.

Ian Osband, Daniel Russo, and Benjamin Van Roy. (more) efficient reinforcement learning via posterior sampling. In *Advances in Neural Information Processing Systems*, pages 3003–3011, 2013.

Florian A Potra and Stephen J Wright. Interior-point methods. *Journal of Computational and Applied Mathematics*, 124(1-2):281–302, 2000.

Martin L Puterman. *Markov decision processes: discrete stochastic dynamic programming*. John Wiley & Sons, 2014.

Shuang Qiu, Xiaohan Wei, Zhuoran Yang, Jieping Ye, and Zhaoran Wang. Upper confidence primal-dual reinforcement learning for cmdp with adversarial loss. In H. Larochelle, M. Ranzato, R. Hadsell, M. F. Balcan, and H. Lin, editors, *Advances in Neural Information Processing Systems*, volume 33, pages 15277–15287. Curran Associates, Inc., 2020. URL https://proceedings.neurips.cc/paper/2020/file/ae95296e27d7f695f891cd26b4f37078-Paper.pdf.

Rahul Singh, Abhishek Gupta, and Ness B Shroff. Learning in markov decision processes under constraints. *arXiv preprint arXiv:2002.12435*, 2020.

Jean Tarbouriech and Alessandro Lazaric. Active exploration in markov decision processes. In *The 22nd International Conference on Artificial Intelligence and Statistics*, pages 974–982. PMLR, 2019.

Christopher J. C. H. Watkins and Peter Dayan. Q-learning. *Machine Learning*, 8(3):279–292, May 1992. ISSN 1573-0565. doi: 10.1007/BF00992698. URL `https://doi.org/10.1007/BF00992698`.

Tengyu Xu, Yingbin Liang, and Guanghui Lan. A primal approach to constrained policy optimization: Global optimality and finite-time analysis. *arXiv preprint arXiv:2011.05869*, 2020.

Liyuan Zheng and Lillian Ratliff. Constrained upper confidence reinforcement learning. In *Learning for Dynamics and Control*, pages 620–629. PMLR, 2020.

[accepted]uai2022

[american]babel bm algorithm algorithmicx algpseudocode mathtools amsmath,amssymb,amsthm,amsfonts,mathrsfs subfigure enumitem

Definition Theorem Assumption Lemma Corollary Proposition Remark Problem Example Claim Observation

natbib   mathtools booktabs tikz

# Regret Guarantees for Model-Based Reinforcement Learning with Long-Term Average Constraints

**Mridul Agarwal**[1]    **Qinbo Bai**[1]    **Vaneet Aggarwal**[2,1]    **Mridul Agarwal**[1]    **Qinbo Bai**[1]    **Vaneet Aggarwal**[2,1]

[1]School of Electrical and Computer Engineering., Purdue University, West Lafayette, Indiana, USA
[2]School of Industrial Engineering., Purdue University, West Lafayette, Indiana, USA
[1]School of Electrical and Computer Engineering., Purdue University, West Lafayette, Indiana, USA
[2]School of Industrial Engineering., Purdue University, West Lafayette, Indiana, USA

## Abstract

We consider the problem of constrained Markov Decision Process (CMDP) where an agent interacts with an ergodic Markov Decision Process. At every interaction, the agent obtains a reward and incurs $K$ costs. The agent aims to maximize the long-term average reward while simultaneously keeping the $K$ long-term average costs lower than a certain threshold. In this paper, we propose CMDP-PSRL, a posterior sampling based algorithm using which the agent can learn optimal policies to interact with the CMDP. We show that with the assumption of slackness, characterized by $\kappa$, the optimization problem is feasible for the sampled MDPs. Further, for MDP with $S$ states, $A$ actions, and mixing time $T_M$, we prove that following CMDP-PSRL algorithm, the agent can bound the regret of not accumulating rewards from an optimal policy by $\tilde{O}(T_M S\sqrt{AT})$. Further, we show that the violations for any of the $K$ constraints is also bounded by $\tilde{O}(T_M S\sqrt{AT})$. To the best of our knowledge, this is the first work that obtains a $\tilde{O}(\sqrt{T})$ regret bounds for ergodic MDPs with long-term average constraints using a posterior sampling method.

## A    INTRODUCTION

Consider a wireless sensor network where the devices aim to update a server with sensor values. At time $t$, the device can choose to send a packet to obtain a reward of 1 unit or to queue the packet and obtain 0 reward. However, communicating a packet results in $p_t$ power consumption. At time $t$, if the wireless channel condition, $s_t$, is weak and the device chooses to send a packet, the resulting instantaneous power consumption, $p_t$, is high. The goal is to send as many packets as possible while keep the average power consumption, $\sum_{t=1}^{T} p_t/T$, within some limit, say $C$. This environment has state $(s_t, q_t)$ as the channel condition and queue length at time $t$. To limit the power consumption, the agent may choose to send packets when the channel condition is good or when the queue length grows beyond a certain threshold. The agent aims to learn the policies in an *online manner* which requires efficiently balancing exploration of state-space and exploitation of the estimated system dynamics [Singh et al., 2020].

Similar to the example above, many applications require to keep some costs low while simultaneously maximizing the rewards [Altman, 1999]. Owing to the importance of this problem, in this paper, we consider the problem of constrained Markov Decision Processes (CMDP). We aim to develop a reinforcement learning algorithm following which an agent can bound the constraint violations and the regret in obtaining the optimal reward to $o(T)$.

The problem setup, where the system dynamics are known, is extensively studied [Altman, 1999]. For a constrained setup, the optimal policy is possibly stochastic [Altman, 1999, Puterman, 2014]. In the domain where the agent learns the system dynamics and aims to learn good policies online, there has been work where to show asymptotic convergence to optimal policies [Gattami et al., 2021], or even provide regret guarantees when the MDP is episodic [Zheng and Ratliff, 2020, Ding et al., 2021]. Recently, [Singh et al., 2020] considered the problem of online optimization of infinite-horizon communicating Markov Decision Processes with long-term average constraints. They provide an optimism based algorithm where confidence bounds on each transition probabilities $p(s'|s, a)$ is constructed. Using this, they obtain a regret bound of

*Accepted for the 38[th] Conference on Uncertainty in Artificial Intelligence* (UAI 2022).

$\tilde{O}\left(\sqrt{SAT} + T_M T^{2/3}\right)$[1]. Additionally, finding the optimistic policy is a computationally intensive task as the number of optimization variables become $S^2 \times A$ for MDP with $S$ states and $A$ actions.

In this paper, we consider the reinforcement learning an infinite-horizon ergodic MDP [Tarbouriech and Lazaric, 2019, Gattami et al., 2021] with long-term average constraints. We use $\ell_1$ deviation bounds [Jaksch et al., 2010] and use a Bellman error analysis to bound the reward regret of the MDP as $\tilde{O}(T_M S\sqrt{AT})$. Additionally, we also bound the constraint violations as $\tilde{O}(T_M S\sqrt{AT})$. We propose a posterior sampling based algorithm where we sample the transition dynamics using a Dirichlet distribution [Osband et al., 2013], which achieves this regret bound.

Unlike optimistic algorithms, the sampled MDP may not be infeasible for the constrained optimization. Hence, we consider slackness characterized by Slater's parameter Ding et al. [2020], which allows us to prove that the optimization problem is feasible even with the sampled MDPs. Posterior sampling also helps to reduces the optimization variables, to find only the optimal policy for the sampled MDP, to only $S \times A$ variables. Finally, we provide numerical examples where the algorithm converges to the calculated optimal policies. To the best of our knowledge, this is the first work to obtain $O(\sqrt{T})$ regret guarantees for the infinite horizon long-term average constraint setup with posterior sampling.

# B    RELATED WORK

Stochastic Optimization using Markov Decision Processes has very rich roots [Howard, 1960]. There have been work in understanding convergence of the algorithm to find optimal policies for known MDPs [Bertsekas and Tsitsiklis, 1996, Altman, 1999]. Also, when the MDP is not known, there are algorithms with asymptotic guarantees for learning the optimal policies [Watkins and Dayan, 1992] which maximize an objective without any constraints. Recent algorithms even achieve finite time near-optimal regret bounds with respect to the number of interactions with the environment [Jaksch et al., 2010, Osband et al., 2013, Agrawal and Jia, 2017, Jin et al., 2018]. Jaksch et al. [2010] uses the optimism principle for minimizing regret for weakly communicating infinite horizon MDPs with bounded diameter. Osband et al. [2013] extended the analysis of Jaksch et al. [2010] to posterior sampling for episodic MDPs and bounded the Bayesian regret and further improved the regret bounds Osband and Van Roy [2017]. Agrawal and Jia [2017] uses a posterior sampling based approach and obtains a frequentist regret for the infinite horizon MDPs with bounded diameter.

In many reinforcement learning settings, the agent not only wants to maximize the rewards but also satisfy certain cost constraints [Altman, 1999]. Early works in this area were pioneered by [Altman and Schwartz, 1991]. They provided an algorithm which combined forced explorations and following policies optimized on empirical estimates to obtain an asymptotic convergence. [Borkar, 2005] studied the constrained RL problem using actor-critic and a two time-scale framework [Borkar, 1997] to obtain asymptotic performance guarantees. Very recently, [Gattami et al., 2021] analyzed the asymptotic performance for Lagrangian based algorithms for infinite-horizon long-term average constraints.

Inspired by the finite-time performance analysis of reinforcement learning algorithm for unconstrained problems, there has been a significant thrust in understanding the finite-time performances of constrained MDP algorithms. [Zheng and Ratliff, 2020] considered an episodic CMDP and use an optimism based algorithm to bound the constraint violation as $\tilde{O}(\sqrt{T^{1.5}})$ with high probability. [Kalagarla et al., 2020] also considered the episodic setup to obtain PAC-style bound for an optimism based algorithm. [Ding et al., 2021] considered the setup of $H$-episode length episodic CMDPs with $d$-dimensional linear function approximation to bound the constraint violations as $\tilde{O}(d\sqrt{H^5T})$ by mixing the optimal policy with an exploration policy. [Efroni et al., 2020] proposes a linear-programming and primal-dual policy optimization algorithm to bound the regret as $O(S\sqrt{H^3T})$. [Qiu et al., 2020] proposed an algorithm which obtains a regret bound of $\tilde{O}(S\sqrt{AH^2T})$ for the problem of adversarial stochastic shortest path. Compared to these works, we focus on setting with infinite horizon long-term average constraints.

After developing a better understanding of the policy gradient algorithms [Agarwal et al., 2020], there has been theoretical work in the area of model-free policy gradient algorithms for constrained MDP and safe reinforcement learning as well. [Xu et al., 2020] consider an infinite horizon discounted setup with constraints and obtain global convergence using policy gradient algorithms. [Ding et al., 2020] also considers an infinite horizon discounted setup. They use a natural policy gradient to update the primal variable and sub-gradient descent to update the dual variable.

Recently [Singh et al., 2020] considered the setup of infinite-horizon CMDPs with long-term average constraints and obtain a regret bound of $\tilde{O}(T^{2/3})$ using an optimism based algorithm and forced explorations. We consider a similar setting with ergodic CMDP and propose a posterior sampling based algorithm to bound the regret as $\tilde{O}(poly(DSA)\sqrt{T})$ using

---

[1]$\tilde{O}(\cdot)$ hides the logarithmic terms

explorations assisted by the ergodicity of the MDP.

## C PROBLEM FORMULATION

We consider an infinite horizon discounted Markov decision process (MDP) $\mathcal{M}$, defined by the tuple $\left(\mathcal{S}, \mathcal{A}, P, r, c_1, \cdots, c^k\right)$. $\mathcal{S}$ denotes a finite set of state space with $|\mathcal{S}| = S$, and $\mathcal{A}$ denotes a finite set of actions with $|\mathcal{A}| = A$. $P : \mathcal{S} \times \mathcal{A} \to \Delta(\mathcal{S})$ denotes the probability $P(s'|s, a)$ of transitioning to state $s'$ from state $s$ after taking action $a$. $r : \mathcal{S} \times \mathcal{A} \to [0, 1]$ denotes the average reward in state $s$ after taking action $a$. $c^k : \mathcal{S} \times \mathcal{A} \to [0, 1]$ denotes average cost incurred by the agent for constraint $k \in [K] = \{1, 2, \cdots, K\}$ after taking action $a$ in state $s$. We use a stochastic policy $\pi : \mathcal{S} \to \Delta(\mathcal{A})$, such that given state $s$, $\pi(a|s)$ is the probability of selecting action $a$.

Note that the a policy $\pi$ induces a Markov chain over the state space of the MDP. Pertaining to the Markov chains generated by the policies for $\mathcal{M}$, we now define the mixing time of MDP.

**Definition 2** (Mixing Time). *Consider the Markov Chain induced by the policy $\pi$ on the MDP $\mathcal{M}$. Let $T^\pi_{s \to s'}$ be a random variable that denotes the first time step when this Markov Chain enters state $s'$ starting from state $s$. Then, the mixing time of the MDP $\mathcal{M}$ is defined as:*

$$T_M = \max_{s' \neq s} \max_{\pi} \mathbb{E}\left[T^\pi_{s \to s'}\right] \tag{37}$$

Similar to Singh et al. [2020], let $P^t_\pi(s)$ be the $t$ step state distribution starting from state $s$ following policy $\pi$ and $P_\pi$ be the steady-state state distribution generated by policy $\pi$.

Our first assumption on the MDP allows any policy to reach any state $s'$ starting from any state $s$, and to converge to a steady state. We formalize the result in the following assumption:

**Assumption 4.** *The MDP $\mathcal{M}$ is ergodic, or $\|P^t_\pi(s) - P_\pi\|_{TV} \leq C\rho^t$ with $P_\pi$ being the long-term steady state distribution induced by policy $\pi$, and $C > 0$ and $\rho < 1$ are problem specific constants. And, the mixing time of the MDP $\mathcal{M}$ is finite or $T_M < \infty$.*

After discussing the transition dynamics of the system, we move to the rewards and costs of the MDP $\mathcal{M}$.

**Assumption 5.** *The reward function $r(s, a)$ and the costs $c^k(s, a), k \in [K]$ are known to the agent.*

We note that in most of the problems, rewards are engineered. Hence, Assumption 2 is justified in many setups. However, the system dynamics are stochastic and typically not known.

Following a policy $\pi$, the expected long-term average cost are given by $\zeta^{P,k}_\pi$. Also, we denote the average long-term reward using $\zeta^{P,k}_\pi$. Formally, we have:

$$\zeta^{P,k}_\pi = \mathbb{E}_{s_0, a_0, s_1, a_1, \cdots}\left[\lim_{\tau \to \infty} \frac{1}{\tau} \sum_{t=0}^{\tau} c^k\left(s_t, a_t\right)\right] \tag{38}$$

$$\lambda^{P,r}_\pi = \mathbb{E}_{s_0, a_0, s_1, a_1, \cdots}\left[\lim_{\tau \to \infty} \frac{1}{\tau} \sum_{t=0}^{\tau} r\left(s_t, a_t\right)\right] \tag{39}$$

$$s_0 \sim \rho_0(s_0), \ a_t \sim \pi(a_t|s_t), \ s_{t+1} \sim P(s_{t+1}|s_t, a_t)$$

For brevity, in the rest of the paper, $\mathbb{E}_{s_t, a_t, s_{t+1}; t \geq 0}[\cdot]$ will be denoted as $\mathbb{E}_{\rho, \pi, P}[\cdot]$, where $s_0 \sim \rho_0(s_0)$, $a_t \sim \pi(s_t|a_t)$, $s_{t+1} \sim P(s_{t+1}|s_t, a_t)$. Both, $\zeta^{P,k}_\pi$ and $\lambda^{P,r}_\pi$ satisfy the following form of Bellman equation:

$$\lambda^{P,r}_\pi + h^{P,r}_\pi(s) = \sum_a \pi(a|s)r(s, a)$$
$$+ \sum_{s'} \sum_a \pi(a|s)P(s'|s, a)h^{P,r}_\pi(s) \tag{40}$$

$$\zeta^{P,k}_\pi + h^{P,r}_\pi(s) = \sum_a \pi(a|s)c^k(s, a)$$
$$+ \sum_{s'} \sum_a \pi(a|s)P(s'|s, a)h^{P,k}_\pi(s) \tag{42}$$

where $h_\pi^{P,r}(s)$ is the bias for reward and $h_\pi^{P,k}$ is the bias for cost $k \in [K]$.

The objective is find a policy $\pi^*$ which is the solution of the following optimization problem.

$$\max_\pi \lambda_\pi^{P,r} \quad \text{s.t.} \tag{43}$$

$$\zeta_\pi^{P,k} \leq C_k \quad \forall\, k \in [K] \tag{44}$$

where $C_k\; \forall\, k \in [K]$ are the bounds on the average costs which the agent needs to satisfy.

After formulating the optimization problem, we now state our next assumption characterizing the slackness.

**Assumption 6.** *There exists a policy $\pi$, and one constant $\kappa \geq 2ST_M\sqrt{14A\log(AT)/\sqrt{T}} + CST_M/((1-\rho)\sqrt{T})$ such that*

$$\zeta_\pi^{P,k} \leq C_k - \kappa \tag{45}$$

The slackness assumption is mild because, in various applications some a priori knowledge about a strictly feasible policy is available. Hence, this assumption is again a standard assumption in the constrained RL literature Efroni et al. [2020], Ding et al. [2021, 2020]. $\kappa$ is referred as Slater's constant. Ding et al. [2021] assumes that the Slater's constant $\kappa$ is known.

Any online algorithm starting with no prior knowledge will require to obtain estimates of transition probabilities $P$ and obtain reward $r$ and costs $c^k, \forall\, k \in [K]$ for each state action pair. Initially, when algorithm does not have good estimates of the model, it accumulates a regret as well as violates constraints as it does not know the optimal policy. We define reward regret $R(T)$ as the difference between the cumulative reward obtained $r_t$ vs the expected rewards from running the optimal policy $\pi^*$ for $T$ steps, or

$$R(T) = T\lambda_{\pi^*}^{P,r} - \sum_{t=1}^{T} r(s_t, a_t) \tag{46}$$

Additionally, we define constraint regret $R_k(T)$ for each constraint $k \in [K]$ as the gap between the cumulative cost incurred $c_t^k, k \in [K]$ and constraint bounds, or

$$R^k(T) = \left( \sum_{t=1}^{T} c^k(s_t, a_t) - TC_k \right)_+, \tag{47}$$

where $(x)_+ = \max(0, x)$.

In the following section, we present a model-based algorithm to obtain this policy $\pi^*$, and reward regret and the constraint regret accumulated by the algorithm.

# D   THE CMDP-PSRL ALGORITHM

For infinite horizon optimization problems (or $\tau \to \infty$), we can use steady state distribution of the state to obtain expected long-term rewards or costs [Puterman, 2014]. We use

$$\zeta_\pi^{P,k} = \sum_{s \in \mathcal{S}} \sum_{a \in \mathcal{A}} c_k(s,a) d_\pi^P(s,a), \;\; \forall\, k \in [K] \tag{48}$$

$$\lambda_\pi^{P,r} = \sum_{s \in \mathcal{S}} \sum_{a \in \mathcal{A}} r(s,a) d_\pi^P(s,a) \tag{49}$$

where $d_\pi^P(s,a)$ is the steady state joint distribution of the state and actions under policy $\pi$.

Based on the above formulation, we can solve the joint optimization problem of following form

$$\max_{d(s,a)} \sum_{s \in \mathcal{S}} \sum_{a \in \mathcal{A}} r(s,a) d(s,a) \tag{50}$$

with the following set of constraints,

$$\sum_{a \in \mathcal{A}} d(s', a) = \sum_{s \in \mathcal{S}, a \in \mathcal{A}} P(s'|s, a) d(s, a) \tag{51}$$

$$\sum_{s \in \mathcal{S}, a \in \mathcal{A}} d(s, a) = 1, \ d(s, a) \geq 0 \tag{52}$$

$$\sum_{s \in \mathcal{S}} \sum_{a \in \mathcal{A}} c_k(s, a) d(s, a) \leq C_k \ \forall \, k \in [K] \tag{53}$$

for all $s' \in \mathcal{S}$, $\forall \, s \in \mathcal{S}$, and $\forall \, a \in \mathcal{A}$. Equation (15) denotes the constraint on the transition structure for the underlying Markov Process. Equation (16) ensures that the solution is a valid probability distribution. Finally, Equation (17) are the constraints for the constrained MDP setup which the policy must satisfy.

Note that arguments in Equation (14) are linear, and the constraints in Equation (15) and Equation (16) are linear, this is a linear programming problem. Since convex optimization problems can be solved in polynomial time [Potra and Wright, 2000], we can use standard approaches to solve Equation (14). After solving the optimization problem, we obtain the optimal policy from the obtained steady state distribution $d^*(s, a)$ as,

$$\pi^*(a|s) = \frac{\mathbb{P}(s, a)}{\mathbb{P}(s)} = \frac{d^*(s, a)}{\sum_{b \in \mathcal{A}} d^*(s, b)} \ \forall \, s \in \mathcal{S} \tag{54}$$

Since we assumed that the CMDP is ergodic, the Markov Chain induced from policy $\pi$ is ergodic. Hence, every state is reachable following the policy $\pi^*$, we have $\mathbb{P}(s) > 0$ and Equation (18) is defined for all states $s \in \mathcal{S}$.

Further, since we assumed that the induced Markov Chain is irreducible for all stationary policies, we assume Dirichlet distribution as prior for the state transition probability $P(s'|s, a)$. Dirichlet distribution is also used as a standard prior in literature [Agrawal and Jia, 2017, Osband et al., 2013]. Further, there exists a steady state distribution when the transition probability is sampled from a Dirichlet distribution [Agarwal et al., 2022, Proposition 1].

The complete constrained posterior sampling based algorithm, which we name CMDP-PSRL, is described in Algorithm 1. The algorithm proceeds in epochs, and a new epoch is started whenever the visitation count in epoch $e$, $\nu_e(s, a)$, is at least the total visitations before episode $e$, $N_e(s, a)$, for any state action pair (Line 8). In Line 9, we sample transition probabilities $\tilde{P}$ using the updated posterior and in Line 10, we update the policy using the optimization problem specified in Equation (14)-(17) for $P = \tilde{P}_e$. Further, if the sampled MDP does not satisfy the cost constraint in Equation (17), we ignore that constraint [2] for that epoch.

# E  ANALYSIS

We first obtain the feasibility of the optimization problem Equation (14)-(17) for the sampled MDP. We note that we assumed slackness in the true MDP with transition probabilities $P$. Hence, if the the sampled MDP is close to the true MDP, the deviation in the cost will be less and there will be a policy which satisfies the constraint in Equation (17). We formalize this intuition in the following result.

**Lemma 2.** *Following Algorithm 1, if $t_{e+1} - t_e \geq \sqrt{T}$ and $\|\tilde{P}_e(\cdot|s, a) - P(\cdot|s, a)\|_1 \leq \sqrt{\frac{14S \log(2At)}{N_e(s, a)}} \forall \, s, a$ there exists a policy $\pi$ which satisfies,*

$$\zeta_\pi^{\tilde{P}_e, k} \leq C_k \ \forall \, k \in [K], \tag{55}$$

*and the optimization problem in Equation (14)-(17) is feasible, where $t_e$ is the start time of epoch $e$.*

*Proof Outline.* We consider the policy $\pi$ which satisfies the Slater's condition in Equation (9). We then consider the Bellman error of taking one step in MDP with transition probabilities $\tilde{P}_e$ and then following policy $\pi$ on the MDP with transition probabilities $P$. Now, using [Agarwal et al., 2022, Lemma 1] relating the average costs following policy $\pi$ with $P$ and $\tilde{P}_e$ ($\zeta_\pi^{P, k}$, and $\zeta_\pi^{\tilde{P}_e, k}$ for all $k \in [K]$ respectively) with the Bellman error gives the required result. The complete proof is provided in the supplementary material. $\qquad \square$

---

[2] We will show in the analysis that cumulative constraint violations are still bounded.

---

**Algorithm 2** CMDP-PSRL

---

1: **Input:** $\mathcal{S}, \mathcal{A}, r, c_1, \cdots, c_K$
2: Initialize $N(s, a, s') = 1 \; \forall (s, a, s') \in \mathcal{S} \times \mathcal{A} \times \mathcal{S}, \; \pi_e(a|s) = \frac{1}{|\mathcal{A}|} \; \forall (a, s) \in \mathcal{A} \times \mathcal{S}, \; e = 0, \; \nu_e(s, a) = N_e(s, a) = 0 \; \forall (s, a) \in$
   $\mathcal{S} \times \mathcal{A}$
3: **for** time index $t = 1, 2, \cdots$ **do**
4:      Observe state $s$
5:      Play action $a \sim \pi(\cdot|s)$
6:      Observe rewards $\{r^k\}$ and next state $s'$
7:      $\nu_e(s, a) + = 1, \; N(s, a, s') + = 1$
8:      **if** $\nu_e(s, a) \geq \max(1, N_e(s, a))$ for any $s, a$ **then**
9:          $\tilde{P}_e(s'|a, s) \sim Dir(N(s, a, s')) \; \forall \, (s, a, s')$
10:          Solve steady state distribution $d(s, a)$ as the solution of the optimization problem in Equations (14-17) for $\tilde{P}_e$.
11:          Obtain optimal policy for next epoch, $e + 1$, $\pi_{e+1}$ as

$$\pi_{e+1}(a|s) = \frac{d(s, a)}{\sum_{a \in \mathcal{A}} d(s, a)}$$

12:          $e = e + 1$
13:          $t_e = t$
14:          $\nu_e(s, a) = 0, N_e(s, a) = \sum_{e'}^{e} \nu_{e'}(s, a) \; \forall (s, a)$
15:      **end if**
16: **end for**

---

After obtaining a feasible policy $\pi_e$ maximizing rewards for the sampled MDP, we now quantify is regret. We note that when optimizing for long-term average rewards and long-term average constraints, we want to simultaneously minimize the reward regret and the constraint regrets. Further, if we know the optimal policy $\pi^*$ before hand, the deviations resulting from the stochasticity of the process can still result in some constraint violations. Also, since we sample a MDP, the policy which is feasible for the MDP may violate constraints on the true MDP. We want to bound this gap between $K$ costs for the two MDPs as well.

We aim to quantify the regret from **(R.1)** deviation of long-term average rewards of the optimal policy because of incorrect knowledge of the MDP ($\lambda_{\pi^*}^{P,r} - \lambda_{\pi_e}^{\tilde{P}_e,r}$), **(R.2)** deviation of the long-term average rewards generated by the optimal policy for the sampled MDP on the sampled MDP and the long-term average rewards generated by the optimal policy for the sampled MDP on the true MDP ($\lambda_{\pi_e}^{\tilde{P}_e,r} - \lambda_{\pi_e}^{P,r}$), and **(R.3)** deviation of the expected rewards from following the optimal policy of the sampled MDP ($\lambda_{\pi_e}^{P,r} - r(s_t, a_t)$).

Similarly, the constraint violations for each $k \in [K]$ are incurred from **(C.1)** deviation of long-term average rewards of the optimal policy because of incorrect knowledge of the MDP ($C_k - \zeta_{\pi_e}^{\tilde{P}_e,k}$), **(C.2)** deviation of the long-term average costs generated by the optimal policy for the sampled MDP on the sampled MDP and the long-term average costs generated by the optimal policy for the sampled MDP on the true MDP ($\zeta_{\pi_e}^{\tilde{P}_e,r} - \lambda_{\pi_e}^{P,r}$), and **(C.3)** deviation of the expected costs from following the optimal policy of the sampled MDP ($\zeta_{\pi_e}^{P,r} - c^k(s_t, a_t)$).

We now prove the regret bounds for Algorithm 1. We first give the high level ideas used in obtaining the bounds on regret. We divide the regret into regret incurred in each epoch $e$. Then, we use the posterior sampling lemma [Osband et al., 2013, Lemma 1] to obtain the equivalence between the long-term average rewards of the true MDP $\mathcal{M}$ and the long-term average rewards for the optimal value of the sampled MDP $\widehat{\mathcal{M}}$. This step allows us to deal with the regret from **(R.1)**. Then we use the Bellman error formulation to relate average rewards for the policy $\pi_e$ on $P$ and $\tilde{P}_e$ [Agarwal et al., 2022]. Combining this with Azuma's concentration inequality for Martingales allows us to bound the regret from **(R.2)** and **(R.3)**.

Bounding constraint violations requires similar considerations for **(C.2)** and **(C.3)**. Further, **(C.1)** becomes zero if Equation (17) is feasible for the sampled MDP. However, if Equation (17) is not feasible, the cost may be as high as 1 ($c^k(s, a) \leq 1 \; \forall \; k \in [K]$). We bound the violations by bounding the time-steps for which the optimal policy for unconstrained optimization runs.

To obtain bounds on the regret, we first note that the total number of epochs, $E$, for which the Algorithm 1 runs is bounded by $O(1 + 2SA + SA \log(T))$ from [Jaksch et al., 2010, Proposition 1].

We formally state the regret bounds and constraint violation bounds in Theorem 1 which we prove rigorously in the supplementary material.

**Theorem 2.** *The expected reward regret* $\mathbb{E}\left[R(T)\right]$*, and the expected constraint regret* $\mathbb{E}\left[R_k(T)\right] \; \forall \, k \in [K]$ *of Algorithm 1 are bounded as*

$$\mathbb{E}\left[R(T)\right] \leq O\left(T_M S \sqrt{AT \log(AT)} + \frac{CS^2 A \log T}{1-\rho}\right)$$

$$\mathbb{E}\left[R^k(T)\right] \leq O\left(T_M S \sqrt{AT \log(AT)} + \frac{CS^2 A \log T}{1-\rho}\right)$$

*Proof Outline.* We break the cumulative regret into the regret incurred in each epoch $e$. This gives us:

$$\mathbb{E}\left[R_T\right] = \mathbb{E}\left[\sum_{e=1}^{E} \sum_{t=t_e}^{t_{e+1}-1} \left(\lambda_{\pi^*}^{P,r} - r(s_t, a_t)\right)\right] \tag{56}$$

$$= \sum_{e=1}^{E} \mathbb{E}\left[\sum_{t=t_e}^{t_{e+1}-1} \left(\lambda_{\pi^*}^{P,r} - r(s_t, a_t)\right)\right] \tag{57}$$

$$= \sum_{e=1}^{E} \mathbb{E}\left[\sum_{t=t_e}^{t_{e+1}-1} \left(\lambda_{\pi_e}^{\tilde{P}_e,r} - r(s_t, a_t)\right)\right] \tag{58}$$

$$= \sum_{e=1}^{E} \mathbb{E}\left[\sum_{t=t_e}^{t_{e+1}-1} \left(\lambda_{\pi_e}^{\tilde{P}_e,r} - \lambda_{\pi_e}^{P,r} + \lambda_{\pi_e}^{P,r} - r(s_t, a_t)\right)\right]$$

$$= \sum_{e=1}^{E} \mathbb{E}\left[\sum_{t=t_e}^{t_{e+1}-1} \left(\lambda_{\pi_e}^{\tilde{P}_e,r} - \lambda_{\pi_e}^{P,r}\right)\right]$$

$$+ \mathbb{E}\left[\sum_{e=1}^{E} \sum_{t=t_e}^{t_{e+1}-1} \left(\lambda_{\pi_e}^{P,r} - r(s_t, a_t)\right)\right] \tag{59}$$

The Equation (22) follows from [Osband et al., 2013, Lemma 1] for regret each each epoch of Equation (21). Proceeding from Equation (23) requires additional consideration. Typical proof techniques to bound regret requires a bounded bias-span $(\max_{s,s'}(h_{\pi}^{\tilde{P}_e,r}(s) - h_{\pi}^{\tilde{P}_e,r}(s')))$ which may be large for the sampled MDP. For this, we consider an MDP for the transition probability $P_e^r$ satisfies

$$\lambda_{\pi_e}^{P_e^r,r} \geq \max_{P' \in \mathcal{P}_{t_e}} \lambda_{\pi_e}^{P',r}, \text{ where} \tag{60}$$

$$\mathcal{P}_{t_e} = \Big\{P' : \|P'(\cdot|s,a) - \bar{P}_{t_e}(\cdot|s,a)\|_1$$

$$\leq \sqrt{\frac{14 S \log(AT)}{N_e(s,a)}}\Big\} \; \forall \, s,a$$

where $\bar{P}_{t_e}(\cdot|s,a)$ is the estimated transition probability given $s, a$ at time $t_e$. We now have,

$$R(T) \leq \sum_{e=1}^{E} \mathbb{E}\left[\sum_{t=t_e}^{t_{e+1}-1} \left(\lambda_{\pi_e}^{P_e^r,r} - \lambda_{\pi_e}^{P,r}\right)\right]$$

$$+ \sum_{e=1}^{E} \mathbb{E}\left[\sum_{t=t_e}^{t_{e+1}-1} \left(\lambda_{\pi_e}^{P,r} - r(s_t, a_t)\right)\right] \tag{61}$$

The first term of Equation (25) is bounded by bounding the expected Bellman error. The second term is converted to a Martingale sequence by conditioning it on the state $s_{t_e}$ and is bounded using the ergodicity of the MDP $\mathcal{M}$ and Azuma's concentration inequality. The complete proof on bounding the regret is provided in the supplementary material.

Regarding the constraint violations, for each $k \in [K]$, we want to bound,

$$\mathbb{E}\left[R^k(T)\right] = \mathbb{E}\left[\left(\sum_{t=1}^{T} c_k(s_t, a_t) - TC_k\right)_+\right] \tag{62}$$

We divide the constraint violation regret into regret over epochs as well. Now, for each epoch, we know that the constraint is satisfied by the policy for the sampled MDP. This allows us to obtain:

$$\mathbb{E}\left[R^k(T)\right] = \mathbb{E}\left[\left(\sum_e \sum_{t=t_e}^{t_{e+1}-1} (c_k(s_t, a_t) - C_k)\right)_+\right] \tag{63}$$

$$= \mathbb{E}\left[\left(\sum_e \sum_{t=t_e}^{t_{e+1}-1} \left(\left(c_k(s_t, a_t) - \zeta_{\pi_e}^{P,k}\right)\right.\right.\right.$$
$$\left.\left.\left. + \left(\zeta_{\pi_e}^{P,k} - \zeta_{\pi_e}^{\tilde{P}_e,k}\right) + \left(\zeta_{\pi_e}^{\tilde{P}_e,k} - C_k\right)\right)\right)_+\right] \tag{64}$$

$$= \mathbb{E}\left[\left(\sum_e \sum_{t=t_e}^{t_{e+1}-1} c_k(s_t, a_t) - \zeta_{\pi_e}^{P,k}\right)_+\right.$$
$$+ \left(\sum_e \sum_{t=t_e}^{t_{e+1}-1} \zeta_{\pi_e}^{P,k} - \zeta_{\pi_e}^{\tilde{P}_e,k}\right)_+$$

$$+ \left(\sum_e \sum_{t=t_e}^{t_{e+1}-1} \zeta_{\pi_e}^{\tilde{P}_e,k} - C_k\right)_+\right] \tag{65}$$

$$= \mathbb{E}\left[\left|\sum_e \sum_{t=t_e}^{t_{e+1}-1} \left(c_k(s_t, a_t) - \zeta_{\pi_e}^{P,k}\right)\right|\right.$$
$$+ \left|\sum_e \sum_{t=t_e}^{t_{e+1}-1} \zeta_{\pi_e}^{P,k} - \zeta_{\pi_e}^{\tilde{P}_e,k}\right|$$
$$+ \left(\sum_e \sum_{t=t_e}^{t_{e+1}-1} \zeta_{\pi_e}^{\tilde{P}_e,k} - C_k\right)_+\right] \tag{66}$$

The first term in Equation (28) denotes the difference between the incurred costs and the expected costs from following policy $\pi_e$. The second term denotes the difference between the expected costs from policy $\pi_e$ on the true MDP and on the sampled MDP. The third terms denotes the violations of the policy $\pi_e$ which would be zero if the policy $\pi_e$ satisfies constraint Eqution (17) for the sampled MDP. Equation (29) is obtained from the fact $\max(0, x + y) \leq \max(0, x) + \max(0, y)$ and Equation (28) is obtained from the fact $\max(0, x) \leq |x|$.

The first and second term in Equation (28) are bounded similar to Equation (23), and we focus our attention to the third term. If the optimization problem in Equation (14)-(17) is feasible, the term $(\zeta_{\pi_e}^{\tilde{P}_e,k} - C_k) \leq 0$ and if the optimization equation is

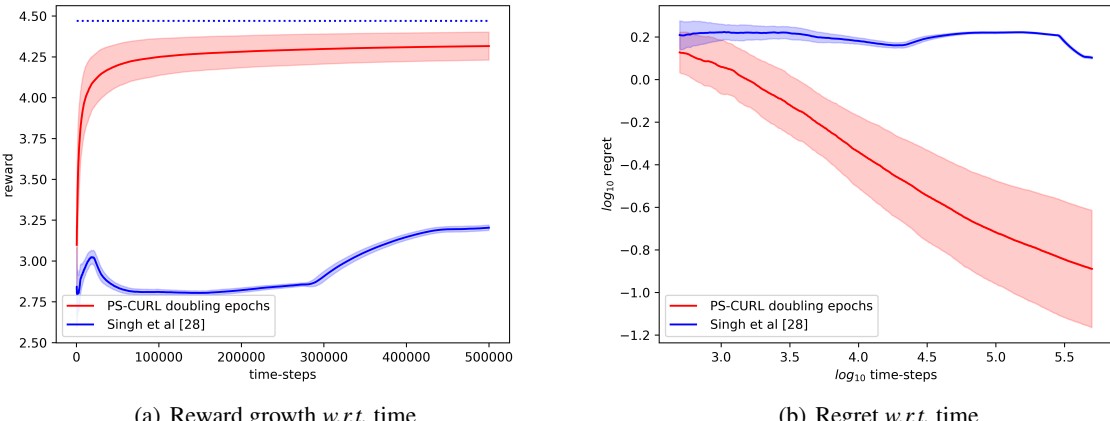

(a) Reward growth *w.r.t.* time          (b) Regret *w.r.t.* time

Figure 3: Reward and regret performance of the proposed CMDP-PSRL algorithm on a flow and service control problem for a single queue. The algorithms is compared against the optimistic algorithm from Singh et al. Singh et al. [2020] compared to which our algorithm extremely well.

infeasible, the term is upper bounded by $1$ as $C_k \geq 0$ and $\zeta_{\pi_e}^{\tilde{P}_e} \leq 1$. Hence, we get:

$$
\left( \sum_e \sum_{t=t_e}^{t_{e+1}-1} \left( \zeta_{\pi_e}^{\tilde{P}_e,k} - C_k \right) \right)_+
$$

$$
\leq \sum_e \left( \sum_{t=t_e}^{t_{e+1}-1} \zeta_{\pi_e}^{\tilde{P}_e,k} - C_k \right)_+ \tag{67}
$$

$$
= \sum_e \left( \sum_{t=t_e}^{t_{e+1}-1} \zeta_{\pi_e}^{\tilde{P}_e,k} - C_k \right)_+ \mathbf{1}\left\{ t_{e+1} - t_e > \sqrt{T} \right\}
$$

$$
+ \sum_e \left( \sum_{t=t_e}^{t_{e+1}-1} \zeta_{\pi_e}^{\tilde{P}_e,k} - C_k \right)_+ \mathbf{1}\left\{ t_{e+1} - t_e \leq \sqrt{T} \right\} \tag{68}
$$

$$
\leq \sum_e \sum_{t=t_e}^{t_{e+1}-1} \mathbf{1}\left\{ t_{e+1} - t_e \leq \sqrt{T} \right\} \tag{69}
$$

$$
\leq \sum_e \sqrt{T} = E\sqrt{T} \tag{70}
$$

$$
\leq (1 + 2SA + SA \log_2(T/SA))\sqrt{T} \tag{71}
$$

where Equation (31) follows from the fact that total violations are less than the cumulative violations are considered per epoch. Equation (33) follows from Lemma 1 as $\left( \zeta_{\pi_e}^{\tilde{P}_e,k} - C_k \right) \leq 0$ when $t_e > \sqrt{T}$ and Equation (35) comes from [Jaksch et al., 2010, Proposition 1]. □

We note that the fundamental setup of unconstrained optimization ($K = 0$), the bound is loose compared to that of UCRL2 algorithm Jaksch et al. [2010]. This is because we use a stochastic policy instead of a deterministic policy. Recall that the optimal policy for CMDP setup is possibly stochastic Altman [1999].

## F  EVALUATION OF THE PROPOSED ALGORITHM

To validate the performance of the proposed CDMP-PSRL algorithm and the understanding of our analysis, we run the simulation on the flow and service control in a single-serve queue, which is introduced in [Altman and Schwartz, 1991]. A

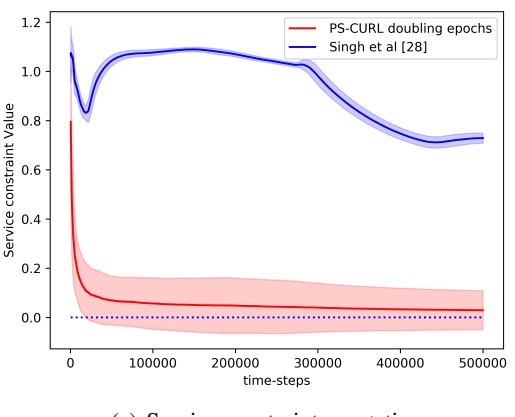

(a) Service constraints *w.r.t.* time

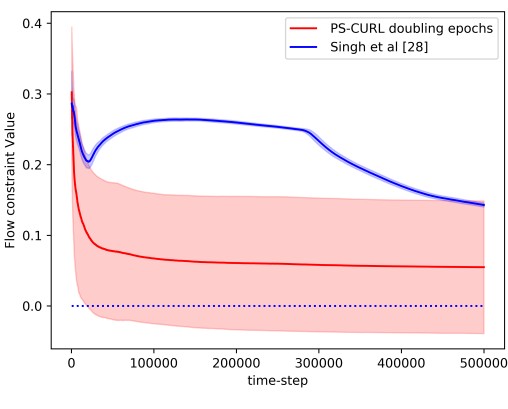

(b) Flow constraints *w.r.t.* time

Figure 4: Constraint violation performance of the proposed CMDP-PSRL algorithm on a flow and service control problem for a single queue. The average constraint violations become zero as the algorithm proceeds, however, it never crosses zero to increase the reward further.

discrete-time single-server queue with a buffer of finite size $L$ is considered in this case. The number of the customer waiting in the queue is considered as the state in this problem and thus $|S| = L + 1$. Two kinds of the actions, service and flow, are considered in the problem and control the number of customers together. The action space for service is a finite subset $A$ in $[a_{min}, a_{max}]$, where $0 < a_{min} \leq a_{max} < 1$. Given a specific service action $a$, the service a customer is successfully finished with the probability $b$. If the service is successful, the length of the queue will reduce by 1. Similarly, the space for flow is also a finite subsection $B$ in $[b_{min}, b_{max}]$. In contrast to the service action, flow action will increase the queue by 1 with probability $b$ if the specific flow action $b$ is given. Also, we assume that there is no customer arriving when the queue is full. The overall action space is the Cartesian product of the $A$ and $B$. According to the service and flow probability, the transition probability can be computed and is given in the Table 1.

Table 2: Transition probability of the queue system

| Current State | $P(x_{t+1} = x_t - 1)$ | $P(x_{t+1} = x_t)$ | $P(x_{t+1} = x_t + 1)$ |
|---|---|---|---|
| $1 \leq x_t \leq L - 1$ | $a(1-b)$ | $ab + (1-a)(1-b)$ | $(1-a)b$ |
| $x_t = L$ | $a$ | $1 - a$ | $0$ |
| $x_t = 0$ | $0$ | $1 - b(1-a)$ | $b(1-a)$ |

For the reward function as $r(s, a, b)$ and the constraints for service and flow as $c^1(s, a, b)$ and $c^2(s, a, b)$, respectively, and stationary policies for service and flow as $\pi_a$ and $\pi_b$, respectively, the problem can be defined as

$$
\max_{\pi_a, \pi_b} \quad \lim_{T \to \infty} \frac{1}{T} \sum_{t=1}^{T} r(s_t, \pi_a(s_t), \pi_b(s_t))
$$

$$
s.t. \quad \lim_{T \to \infty} \frac{1}{T} \sum_{t=1}^{T} c^1(s_t, \pi_a(s_t), \pi_b(s_t)) \geq 0 \tag{72}
$$

$$
\lim_{T \to \infty} \frac{1}{T} \sum_{t=1}^{T} c^2(s_t, \pi_a(s_t), \pi_b(s_t)) \geq 0
$$

According to the discussion in [Altman and Schwartz, 1991], we define the reward function as $r(s, a, b) = 5 - s$, which is an decreasing function only dependent on the state. It is reasonable to give higher reward when the number of customer waiting in the queue is small. For the constraint function, we define $c^1(s, a, b) = -10a + 6$ and $c^2 = -8 * (1 - b)^2 + 2$,

which are dependent only on service and flow action, respectively. Higher constraint value is given if the probability for the service and flow are low and high, respectively.

In the simulation, the length of the buffer is set as $L = 5$. The service action space is set as $[0.2, 0.4, 0.6, 0.8]$ and the flow action space is set as $[0.4, 0.5, 0.6, 0.7]$. We use the length of horizon $T = 50000$ and run 50 independent simulations of the proposed CMDP-PSRL algorithm. We also plot the standard deviation around the mean value in the shadow to show the random error. In order to compare this result to the optimal, we assume that the full information of the transition dynamics is known and then use Linear Programming to solve the problem. The optimal cumulative reward from LP is shown to be $4.47$. The reward performance of the CMDP-PSRL algorithm is shown in the Figure 1 where we observe that the reward converges towards the optimal value. We also plot the constraint violations in Figure 2. The service and flow constraints converge to 0 as expected. We note that the reward of the proposed CMDP-PSRL algorithm becomes closer the optimal reward as the algorithm proceeds, and to further increase the reward, it does not violates the constraint.

We also compared our algorithm against the optimistic algorithm of Singh et al. [2020]. We note that their algorithm performs significantly worse compared to our algorithm. We account this poor performance on two accounts. An optimistic algorithm does not find a policy for transition probabilities close to $P$ for significantly large time. The other issue is because they consider confidence interval for each $P(s'|s, a)$. This also shows in their analysis and hence they obtain a $O(T^{2/3})$ regret bound. Further, the optimization problem takes a significantly more time to solve for optimistic setup. However, the variance of their optimistic algorithm is significantly lower compared to the variance of our CMDP-PSRL algorithm.

# G   CONCLUSION

This paper, considers the setup of reinforcement learning in ergodic infinite-horizon constrained Markov Decision Processes with $K$ long-term average constraint. We propose a posterior sampling based algorithm, CMDP-PSRL, which proceeds in epochs. At every epoch, we sample a new CMDP and generate a solution for the constraint optimization problem. A major advantage of the posterior sampling based algorithm over an optimistic approach is, that it reduces the complexity of solving for the optimal solution of the constraint problem. We also study the proposed CMDP-PSRL algorithm from regret perspective. We bound the regret of the reward collected by the CMDP-PSRL algorithm as $\tilde{O}(T_M S \sqrt{AT} + CS^2 A/(1-\rho))$. Further, we bound the gap between the long-term average costs of the sampled MDP and the true MDP to bound the $K$ constraint violations as $\tilde{O}(T_M S \sqrt{AT} + CS^2 A/(1-\rho))$. Finally, we evaluate the proposed CMDP-PSRL algorithm on a flow control problem for single queue and show that the proposed algorithm performs empirically well. This paper is the first work which obtains a $\tilde{O}(\sqrt{T})$ regret bounds for ergodic MDPs with long-term average constraints using a posterior sampling algorithm. A model-free algorithm that obtains similar regret bounds for infinite horizon long-term average constraints remains an open problem.

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

# A    PROOF FOR REGRET BOUNDS

We now complete the proof of Theorem 1 here.

## A.1    VARIABLE DEFINITIONS

We first define some important variables required for the proof.

We define value $V_{\gamma,\pi}^{P,r}, V_{\gamma,\pi}^{P,k}$ function for rewards $r$ and cost $c^k$ as:

$$V_{\gamma,\pi}^{P,r}(s) = \mathbb{E}\left[\sum_{t=0}^{\infty}\gamma^t r(s_t, a_t)|s_0 = s\right] \tag{73}$$

$$V_{\gamma,\pi}^{P,k}(s) = \mathbb{E}\left[\sum_{t=0}^{\infty}\gamma^t c^k(s_t, a_t)|s_0 = s\right] \tag{74}$$

We also define Q-value $Q_{\gamma,\pi}^{P,r}, Q_{\gamma,\pi}^{P,k}$ function for rewards $r$ and cost $c^k$ as:

$$Q_{\gamma,\pi}^{P,r}(s, a) = \mathbb{E}\left[\sum_{t=0}^{\infty}\gamma^t r(s_t, a_t)|s_0 = s, a_0 = a\right] \tag{75}$$

$$Q_{\gamma,\pi}^{P,k}(s, a) = \mathbb{E}\left[\sum_{t=0}^{\infty}\gamma^t c^k(s_t, a_t)|s_0 = s, a_0 = a\right] \tag{76}$$

Based on this, we define Bellman error $B_{\pi}^{P',r}, B_{\pi}^{P',k}$ function for rewards $r$ and cost $c^k$ as:

$$B_{\pi}^{P',r} = \lim_{\gamma\to 1}\left(r(s, a) + \sum_{s'}P'(s'|s, a)V_{\gamma,\pi}^{P,r}(s, a) - Q_{\gamma,\pi}^{P,r}(s, a)\right) \tag{77}$$

$$B_{\pi}^{P',k} = \lim_{\gamma\to 1}\left(c^k(s, a) + \sum_{s'}P'(s'|s, a)V_{\gamma,\pi}^{P,k}(s, a) - Q_{\gamma,\pi}^{P,k}(s, a)\right) \tag{78}$$

## A.2 AUXILIARY LEMMAS

We now state and prove various lemmas required to complete the proof of Theorem 1.

The first lemma obtains concentration bounds for the sampled MDP. We have:

**Lemma 3.** *The probability that the event*

$$\mathcal{E}_t = \left\{ \|\bar{P}_t(\cdot|s,a) - P(\cdot|s,a)\|_1 \le \sqrt{\frac{14S\log(2AT)}{\max\{1, n_t(s,a)\}}} \forall (s,a) \in \mathcal{S} \times \mathcal{A} \right\} \tag{79}$$

*fails to occur for any $t \le T$ is bounded by $\frac{1}{T^5}$.*

*Proof Outline.* From the result of **?**, the $\ell_1$ distance of a probability distribution over $S$ events with $n$ samples is bounded as:

$$\mathbb{P}\left(\|P(\cdot|s,a) - \bar{P}_t(\cdot|s,a)\|_1 \ge \epsilon\right) \le (2^S - 2)\exp\left(-\frac{n(s,a)\epsilon^2}{2}\right)$$

$$\le (2^S)\exp\left(-\frac{n(s,a)\epsilon^2}{2}\right) \tag{80}$$

Thus, for $\epsilon = \sqrt{\frac{2}{n(s,a)}\log(2^S 20SAT^7)} \le \sqrt{\frac{14S}{n(s,a)}\log(2AT)} \le \sqrt{\frac{14S}{n(s,a)}\log(2AT)}$, we have

$$\mathbb{P}\left(\|P(\cdot|s,a) - \bar{P}_t(\cdot|s,a)\|_1 \ge \sqrt{\frac{14S}{n(s,a)}\log(2AT)}\right) \le (2^S)\exp\left(-\frac{n(s,a)}{2}\frac{2}{n(s,a)}\log(2^S 20SAT^7)\right) \tag{81}$$

$$= 2^S \frac{1}{2^S 20SAT^7} \tag{82}$$

$$= \frac{1}{20AST^7} \tag{83}$$

We sum over the all the possible values of $n(s,a)$ till $t$ time-step to bound the probability that the event $\mathcal{E}_t$ does not occur as:

$$\sum_{n(s,a)=1}^{t} \frac{1}{20SAT^7} \le \frac{1}{20SAT^6} \tag{84}$$

Finally, summing over all the $s, a$, we get

$$\mathbb{P}\left(\|P(\cdot|s,a) - \bar{P}_t(\cdot|s,a)\|_1 \ge \sqrt{\frac{14S}{n(s,a)}\log(2AT)} \forall s,a\right) \le \frac{1}{20t^6} \tag{85}$$

Further, using union bounds and summing over all the $t \le T$, we get

$$\mathbb{P}\left(\|P(\cdot|s,a) - \bar{P}_t(\cdot|s,a)\|_1 \ge \sqrt{\frac{14S}{n(s,a)}\log(2AT)} \forall s,a \forall t \le T\right) \le \sum_{t=1}^{T} \frac{1}{20T^6} \tag{86}$$

$$\le \frac{1}{T^5} \tag{87}$$

$\square$

The next lemma relates the difference between average per step reward $\lambda_\pi^{P,r}$ (or cost $\lambda_\pi^{P,k}$) for following policy $\pi$ on true MDP with transition probabilities and average per step reward $\lambda_\pi^{\tilde{P},r}$ for following policy $\pi$ on MDP with transition probabilities $\tilde{P}$ with the Bellman error $B_\pi^{\tilde{P},r}(s,a)$ as:

**Lemma 4.** *The difference of long-term average rewards for running the policy $\pi$ on the MDP, $\lambda_\pi^{\tilde{P},r}$, and the average long-term average rewards for running the policy $\pi$ on the true MDP, $\lambda_\pi^{\tilde{P},r}$, is the long-term average Bellman error as*

$$\lambda_\pi^{\tilde{P},r} - \lambda_\pi^{P,r} = \sum_{s,a} d_\pi(s,a) B_\pi^{\tilde{P},r}(s,a) = \mathbb{E}_{\pi,P}\left[B_\pi^{\tilde{P},r}(s,a)\right]. \tag{88}$$

*Proof.* Note that for all $s \in \mathcal{S}$, we have:

$$V_{\gamma,\pi}^{\tilde{P},r}(s) = \mathbb{E}_{a\sim\pi}\left[Q_{\gamma,\pi}^{\tilde{P},r}(s,a)\right] \tag{89}$$

$$= \mathbb{E}_{a\sim\pi}\left[B_{\gamma,\pi}^{\tilde{P},r}(s,a) + r(s,a) + \gamma \sum_{s'\in\mathcal{S}} P(s'|s,a)V_{\gamma\pi}^{\tilde{P},r}(s')\right] \tag{90}$$

where Equation (54) follows from the definition of the Bellman error for state action pair $s, a$.

Similarly, for the true MDP, we have,

$$V_{\gamma,\pi}^{P,r}(s) = \mathbb{E}_{a\sim\pi}\left[Q_{\gamma,\pi}^{P,r}(s,a)\right] \tag{91}$$

$$= \mathbb{E}_{a\sim\pi}\left[r(s,a) + \gamma \sum_{s'\in\mathcal{S}} P(s'|s,a)V_{\gamma,\pi}^{P,r}(s')\right] \tag{92}$$

Subtracting Equation (56) from Equation (54), we get:

$$V_{\gamma,\pi}^{\tilde{P},r}(s) - V_{\gamma,\pi}^{P,r}(s) = \mathbb{E}_{a\sim\pi}\left[B_{\gamma,\pi}^{\tilde{P},r}(s,a) + \gamma \sum_{s'\in\mathcal{S}} P(s'|s,a)\left(V_{\gamma,\pi}^{\tilde{P},r} - V_{\gamma,\pi}^{P,r}\right)(s')\right] \tag{93}$$

$$= \mathbb{E}_{a\sim\pi}\left[B_{\gamma,\pi}^{\tilde{P},r}(s,a)\right] + \gamma \sum_{s'\in\mathcal{S}} P_\pi\left(V_{\gamma,\pi}^{\tilde{P},r} - V_{\gamma,\pi}^{P,r}\right)(s') \tag{94}$$

Using the vector format for the value functions, we have,

$$\bar{V}_{\gamma,\pi}^{\tilde{P},r} - \bar{V}_{\gamma,\pi}^{P,r} = (I - \gamma P_\pi)^{-1} B_{\gamma,\pi}^{P,r} \tag{95}$$

Now, converting the value function to average per-step reward we have,

$$\lambda_\pi^{\tilde{P},r}\mathbf{1}_S - \lambda_\pi^{P,r}\mathbf{1}_S = \lim_{\gamma\to1}(1-\gamma)\left(\bar{V}_{\gamma,\pi}^{\tilde{P},r} - \bar{V}_{\gamma,\pi}^{P,r}\right) \tag{96}$$

$$= \lim_{\gamma\to1}(1-\gamma)(I - \gamma P_\pi)^{-1} B_{\gamma,\pi}^{\tilde{P},r} \tag{97}$$

$$= \left(\sum_{s,a} d_\pi^P(s,a) B_\pi^{\tilde{P},r}(s,a)\right)\mathbf{1}_S \tag{98}$$

where the last equation follows from the definition of occupancy measures by Puterman [2014], and the existence of the limit $\lim_{\gamma\to1} B_{\gamma,\pi}^{\tilde{P},r}$ in Equation (72). $\square$

After relating the gap between the long-term average rewards of policy $\pi_e$ on the two MDPs, we now want to bound the sum of Bellman error over an epoch. For this, we first bound the Bellman error for a particular state action pair $s, a$ in the form of following lemma. We have,

**Lemma 5.** *For an MDP with rewards $r(s,a)$ and transition probability $\tilde{P}(s'|s,a)$ such that $\|\tilde{P}(\cdot|s,a) - P(\cdot|s,a)\|_1 \le \epsilon_{s,a}$, the Bellman error $B_{\pi_e}^{\tilde{P},r}(s,a)$ for state-action pair $s, a$ is upper bounded as*

$$B_\pi^{\tilde{P},r}(s,a) \le \left\|\tilde{P}(\cdot|s,a) - P(\cdot|s,a)\right\|_1 \|h_\pi^{\tilde{P},r}(\cdot)\|_\infty \tag{99}$$

*where $\|h_\pi^{\tilde{P},r}(\cdot)\|_\infty$ is the bias-span of the MDP with transition probability $\tilde{P}$.*

*Proof.* Starting with the definition of Bellman error in Equation (41), we get

$$B_\pi^{\tilde{P},r}(s,a) = \lim_{\gamma \to 1} B_{\gamma,\pi}^{\tilde{P},r}(s,a) \tag{100}$$

$$= \lim_{\gamma \to 1} \left( Q_{\gamma,\pi}^{\tilde{P},r}(s,a) - \left( r(s,a) + \gamma \sum_{s' \in \mathcal{S}} P(s'|s,a) V_{\gamma,\pi}^{\tilde{P},r} \right) \right) \tag{101}$$

$$= \lim_{\gamma \to 1} \left( \left( r(s,a) + \gamma \sum_{s' \in \mathcal{S}} \tilde{P}(s'|s,a) V_{\gamma,\pi}^{\tilde{P},r}(s') \right) - \left( r(s,a) + \gamma \sum_{s' \in \mathcal{S}} P(s'|s,a) V_{\gamma,\pi}^{\tilde{P},r}(s') \right) \right) \tag{102}$$

$$= \lim_{\gamma \to 1} \gamma \sum_{s' \in \mathcal{S}} \left( \tilde{P}(s'|s,a) - P(s'|s,a) \right) V_{\gamma,\pi}^{\tilde{P},r}(s') \tag{103}$$

$$= \lim_{\gamma \to 1} \gamma \left( \sum_{s' \in \mathcal{S}} \left( \tilde{P}(s'|s,a) - P(s'|s,a) \right) V_{\gamma,\pi}^{\tilde{P},r}(s') + V_{\gamma,\pi}^{\tilde{P},r}(s) - V_{\gamma,\pi}^{\tilde{P},r}(s) \right) \tag{104}$$

$$= \lim_{\gamma \to 1} \gamma \Big( \sum_{s' \in \mathcal{S}} \left( \tilde{P}(s'|s,a) - P(s'|s,a) \right) V_{\gamma,\pi}^{\tilde{P},r}(s') - \sum_{s' \in \mathcal{S}} \tilde{P}(s'|s,a) V_{\gamma,\pi}^{\tilde{P},r}(s)$$

$$+ \sum_{s' \in \mathcal{S}} P(s'|s,a) V_{\gamma,\pi}^{\tilde{P},r}(s) \Big) \tag{105}$$

$$= \lim_{\gamma \to 1} \gamma \left( \sum_{s' \in \mathcal{S}} \left( \tilde{P}(s'|s,a) - P(s'|s,a) \right) \left( V_{\gamma,\pi}^{\tilde{P},r}(s') - V_{\gamma,\pi}^{\tilde{P},r}(s) \right) \right) \tag{106}$$

$$= \left( \sum_{s' \in \mathcal{S}} \left( \tilde{P}(s'|s,a) - P(s'|s,a) \right) \lim_{\gamma \to 1} \gamma \left( V_{\gamma,\pi}^{\tilde{P},r}(s') - V_{\gamma,\pi}^{\tilde{P},r}(s) \right) \right) \tag{107}$$

$$= \left( \sum_{s' \in \mathcal{S}} \left( \tilde{P}(s'|s,a) - P(s'|s,a) \right) h_\pi^{\tilde{P},r}(s') \right) \tag{108}$$

$$\leq \left\| \tilde{P}(\cdot|s,a) - P(\cdot|s,a) \right\|_1 \| h_\pi^{\tilde{P},r}(\cdot) \|_\infty \tag{109}$$

$$\leq \epsilon_{s,a} \tilde{T}_M \tag{110}$$

where Equation (67) comes from the assumption that the rewards are known to the agent. Equation (71) follows from the fact that the difference between value function at two states is bounded. Equation (72) comes from the definition of bias term Puterman [2014] where $h$ is the bias of the policy $\pi$ when run on the sampled MDP. Equation (73) follows from Hölder's inequality. In Equation (74), the $\ell_1$ norm of probability vector difference is bounded from the definition.

Additionally, note that the $\ell_1$ norm in Equation (73) is bounded by 2. Thus the Bellman error is loose upper bounded by $2\| h_\pi^{\tilde{P},r}(\cdot) \|_\infty$ for all state-action pairs. □

**Lemma 6** (Bounded Span of optimal MDP in confidence interval). *For a MDP with rewards $r(s,a)$ and transition probabilities $P_e^r = \arg\max_{P_e \in \mathcal{P}_{t_e}} \lambda_{\pi_e}^{P_e,r}$, for policy $\pi_e$, the difference of bias of any two states $s$, and $s'$, is upper bounded by the mixing time of the true MDP $T_M$ as:*

$$h_{\pi_e}^{P_e^r,r}(s) - h_{\pi_e}^{P_e^r,r}(s') \leq T_M \ \forall \ s, s' \in \mathcal{S} \tag{111}$$

*Proof.* Note that $\lambda_{\pi_e}^{P_e^r,r} \geq \lambda_{\pi_e}^{P',r}$ for all $P' \in \mathcal{P}_{t_e}$. Now, consider the following Bellman equation:

$$h_{\pi_e}^{P_e^r,r}(s) = r_{\pi_e}(s,a) - \lambda_{\pi_e}^{P_e^r,r} + < P_{\pi_e,e}^r(\cdot|s), h_{\pi_e}^{P_e^r,r} >$$

$$= T h_{\pi_e}^{P_e^r,r}(s) \tag{112}$$

where $r_{\pi_e}(s) = \sum_a \pi_e(a|s) r(s,a)$ and $P_{\pi_e,e}^r(s'|s) = \sum_a \pi(a|s) P_e^r(s'|s,a)$.

Consider two states $s, s' \in \mathcal{S}$. Also, let $\tau$ be a random variable defined as:

$$\tau = \min\{t \geq 1 : s_t = s', s_1 = s\} \tag{113}$$

We also define another operator,

$$\bar{T}h(s) = \begin{cases} \min_{s,a} r(s,a) - \lambda_{\pi_e}^{P^r_e, r} + < P_{\pi_e}(\cdot|s), h >, & s \neq s' \\ h_{\pi_e}^{P^r_e, r}(s'), & s = s' \end{cases} \tag{114}$$

where $P_{\pi_e}(\cdot|s) = \sum_a \pi_e(a|s) P(s'|s,a)$.

Now, note that

$$h(s) = Th(s) \tag{115}$$

$$= \max_{P' \in \mathcal{P}_{t_e}} \left( r_{\pi_e}^r(s) - \lambda_{\pi_e}^{P^r_e, r} + < P'_{\pi_e}, h > \right) \tag{116}$$

$$\geq r_{\pi_e}^r(s) - \lambda_{\pi_e}^{P^r_e, r} + < P_{\pi_e}, h > \tag{117}$$

$$\geq \min_{s,a} r(s,a) - \lambda_{\pi_e}^{P^r_e, r} + < P_{\pi_e}, h > \tag{118}$$

$$= \bar{T}h(s) \tag{119}$$

Further, for any two vectors $u, v$, where all the elements of $u$ are not smaller than $w$ we have $\bar{T}u \geq \bar{T}w$. Hence, we have $\bar{T}^n h_\pi^{P,r}(s) \leq h_\pi^{P,r}(s)$ for all $s$. Unrolling the recurrence, we have

$$h_\pi^{P^r,r}(s) \geq \bar{T}^n h_\pi^{P^r,r}(s) = \mathbb{E}\left[ -(\lambda_\pi^{P^r,r} - \min_{s,a} r(s,a))(n \wedge \tau) + h_\pi^{P^r,r}(s_{n \wedge \tau}) \right] \tag{120}$$

For $\lim n \to \infty$, we have $h_\pi^{P^r_e, r}(s) \geq h_\pi^{P^r_e, r}(s') - T_M$, completing the proof. □

## A.3 PROOF OF RESULTS FROM MAIN TEXT

After stating the necessary lemmas, we can now prove Lemma 1 and Theorem 1.

*Proof of Theorem 1.* We continue our proof from Equation (25). We had:

$$R(T) \leq \sum_{e=1}^{E} \mathbb{E}\left[ \sum_{t=t_e}^{t_{e+1}-1} \left( \lambda_{\pi_e}^{P^r_e, r} - \lambda_{\pi_e}^{P, r} \right) \right] + \sum_{e=1}^{E} \mathbb{E}\left[ \sum_{t=t_e}^{t_{e+1}-1} \left( \lambda_{\pi_e}^{P, r} - r(s_t, a_t) \right) \right] \tag{121}$$

$$= R_1(T) + R_2(T) \tag{122}$$

where $R_1(T)$ and $R_2(T)$ are:

$$R_1(T) = \sum_{e=1}^{E} \mathbb{E}\left[ \sum_{t=t_e}^{t_{e+1}-1} \left( \lambda_{\pi_e}^{P^r_e, r} - \lambda_{\pi_e}^{P, r} \right) \right] \tag{123}$$

$$R_2(T) = \sum_{e=1}^{E} \mathbb{E}\left[ \sum_{t=t_e}^{t_{e+1}-1} \left( \lambda_{\pi_e}^{P, r} - r(s_t, a_t) \right) \right] \tag{124}$$

We first consider $R_2(T)$ term. We start by defining filtration $\mathcal{H}_t = \{s_0, a_0, \cdots, s_t, a_t\}$ as the set of of observed states and played actions. Further, we have $\lambda_{\pi_e}^{P, r}$ as

$$\lambda_{\pi_e}^{P, r} = \mathbb{E}_{(s,a) \sim \pi_e, P}[r(s,a)] \tag{125}$$

We have,

$$\mathbb{E}_{(s,a) \sim \pi_e, P}[r(s,a)] = \mathbb{E}_{(s,a) \sim \pi_e, P}[r(s,a)] \pm \mathbb{E}_{(s_t,a_t) \sim \pi_e, P}[r(s_t, a_t)|\mathcal{H}_{t_e-1}] \tag{126}$$

$$= \mathbb{E}_{(s_t,a_t) \sim \pi_e, P}[r(s_t, a_t)|\mathcal{H}_{t_e-1}] + \left( \mathbb{E}_{(s,a) \sim \pi_e, P}[r(s,a)] - \mathbb{E}_{(s_t,a_t) \sim \pi_e, P}[r(s_t, a_t)|\mathcal{H}_{t_e-1}] \right) \tag{127}$$

$$\leq \mathbb{E}_{(s_t,a_t) \sim \pi_e, P}[t(s_t, a_t)|\mathcal{H}_{t_e-1}] + 2 \left( \|\pi_e(a|s) d_{\pi_e}(s) - \pi_e(a|s) P_{\pi, s_{t_e-1}}^{t-t_e+1}(s)\|_{TV} \right) \tag{128}$$

$$\leq \mathbb{E}_{(s_t,a_t) \sim \pi_e, P}[r(s_t, a_t)|\mathcal{H}_{t_e-1}] + 2CS\rho^{t-t_e} \tag{129}$$

Hence, we have,

$$\sum_{t=t_e}^{t_{e+1}-1} \left( \lambda_{\pi_e}^{P,r} - r(s_t, a_t) \right) = \sum_{t=t_e}^{t_{e+1}-1} \left( \mathbb{E}_{(s,a)\sim\pi_e,P}[r(s,a)] - r(s_t, a_t) \right) \tag{130}$$

$$\leq \sum_{t=t_e}^{t_{e+1}-1} \left( \mathbb{E}_{(s_t,a_t)\sim\pi_e,P}[r(s_t, a_t)|\mathcal{H}_{t_e-1}] + 2CS\rho^{t-t_e} - r(s_t, a_t) \right) \tag{131}$$

$$\leq \sum_{t=t_e}^{t_{e+1}-1} \left( \mathbb{E}_{(s_t,a_t)\sim\pi_e,P}[r(s_t, a_t)|\mathcal{H}_{t_e-1}] - r(s_t, a_t) \right) + \sum_{t=t_e}^{\infty} 2CS\rho^{t-t_e} \tag{132}$$

$$\leq \sum_{t=t_e}^{t_{e+1}-1} \left( \mathbb{E}_{(s_t,a_t)\sim\pi_e,P}[r(s_t, a_t)|\mathcal{H}_{t_e-1}] - r(s_t, a_t) \right) + \frac{2CS}{1-\rho} \tag{133}$$

Using Azuma-Hoeffding's inequality, we get,

$$\sum_{t=t_e}^{t_{e+1}-1} \left( \mathbb{E}_{(s_t,a_t)\sim\pi_e,P}[r(s_t, a_t)|\mathcal{H}_{t_e-1}] - r(s_t, a_t) \right) \leq 2\sqrt{(t_{e+1} - t_e)\log(2T)} \tag{134}$$

with probability at least $1 - 1/T$. Summing over all the epochs and using Cauchy-Schwarz inequality, we get:

$$\sum_{t=t_e}^{t_{e+1}-1} \left( \lambda_{\pi_e}^{P,r} - r(s_t, a_t) \right) = \sum_{e=1}^{E} \left( \sum_{t=t_e}^{t_{e+1}-1} \left( \mathbb{E}_{(s_t,a_t)\sim\pi_e,P}[r(s_t, a_t)|\mathcal{H}_{t_e-1}] - r(s_t, a_t) \right) + \frac{2CS}{1-\rho} \right) \tag{135}$$

$$\leq \sum_{e=1}^{E} 2\sqrt{(t_{e+1} - t_e)\log(2T)} + \frac{2CSE}{1-\rho} \tag{136}$$

$$\leq 2\sqrt{E \sum_{e=1}^{E} (t_{e+1} - t_e)\log(2T)} + \frac{2CSE}{1-\rho} \tag{137}$$

$$= 2\sqrt{ET\log(2T)} + \frac{2CSE}{1-\rho} \tag{138}$$

with probability at least $1 - E/T$. Further, the maximum value of the sum is bounded by $T$ and that event occurs with probability less than $1/T$ which gives,

$$\mathbb{E}[R_2(T)] = \sum_{e=1}^{E} \mathbb{E}\left[ \sum_{t=t_e}^{t_{e+1}-1} \left( \lambda_{\pi_e}^{P,r} - r(s_t, a_t) \right) \right] \tag{139}$$

$$\leq 4\sqrt{T\log(2T)} + \frac{2CSE}{1-\rho} + \frac{E}{T}T \tag{140}$$

$$= E + 4\sqrt{ET\log(2T)} + \frac{2CSE}{1-\rho} \tag{141}$$

We can now focus on the $R_1(T)$ term. We have:

$$R_1(T) = \sum_{e=1}^{T} \mathbb{E}\left[ \sum_{t=t_e}^{t_{e+1}-1} \left( \lambda_{\pi_e}^{P_e^r,r} - \lambda_{\pi_e}^{P,r} \right) \right] \tag{142}$$

$$= \sum_{e=1}^{T} \mathbb{E}\left[ \sum_{t=t_e}^{t_{e+1}-1} \mathbb{E}_{s,a\sim\pi,P}\left[ B_{\pi_e}^{P_e^r,r}(s,a) \right] \right] \tag{143}$$

Similar to Equations (90)-(93), we have:

$$\sum_{e=1}^{E} \sum_{t=t_e}^{t_{e+1}-1} \mathbb{E}_{s,a\sim\pi,P}\left[ B_{\pi_e}^{P_e^r,r}(s,a) \right] \leq \sum_{e=1}^{E} \sum_{t=t_e}^{t_{e+1}-1} \mathbb{E}_{s,a\sim\pi,P}\left[ B_{\pi_e}^{P_e^r,r}(s,a)|\mathcal{H}_{t_e-1} \right] + \sum_{e=1}^{E} \sum_{t=t_e}^{t_{e+1}-1} 2CT_M S\rho^{t-t_e} \tag{144}$$

Again, using Azuma-Hoeffding's inequality, with probability at least $1 - 1/T$ we have:

$$\sum_{t=t_e}^{t_{e+1}-1} \mathbb{E}_{s,a\sim\pi,P}\left[B_{\pi_e}^{P_e^r,r}(s,a)|\mathcal{H}_{t_e-1}\right] \leq \sum_{t=t_e}^{t_{e+1}-1} B_{\pi_e}^{P_e^r,e}(s_t,a_t) + 2T_M\sqrt{(t_{e+1}-t_e)\log(2T)} \tag{145}$$

Summing over all the epochs, we get, with probability at least $1 - E/T$:

$$\sum_{e=1}^{E}\sum_{t=t_e}^{t_{e+1}-1} \mathbb{E}_{s,a\sim\pi,P}\left[B_{\pi_e}^{P_e^r,r}(s,a)|\mathcal{H}_{t_e-1}\right] \leq \sum_{e=1}^{E}\sum_{t=t_e}^{t_{e+1}-1} B_{\pi_e}^{P_e^r,e}(s_t,a_t) + \sum_{e=1}^{E}\sum_{t=t_e}^{t_{e+1}-1} 2T_M\sqrt{(t_{e+1}-t_e)\log(2T)} \tag{146}$$

$$\leq \sum_{e=1}^{E}\sum_{t=t_e}^{t_{e+1}-1} B_{\pi_e}^{P_e^r,e}(s_t,a_t) + 2T_M\sqrt{E\sum_{e=1}^{E}(t_{e+1}-t_e)\log(2T)} \tag{147}$$

$$= \sum_{e=1}^{E}\sum_{t=t_e}^{t_{e+1}-1} B_{\pi_e}^{P_e^r,e}(s_t,a_t) + 2T_M\sqrt{ET\log(2T)} \tag{148}$$

$$\leq \sum_{e=1}^{E}\sum_{t=t_e}^{t_{e+1}-1} \left\|\tilde{P}(\cdot|s,a)-P(\cdot|s,a)\right\|_1 \|h_\pi^{\tilde{P},r}(\cdot)\|_\infty + 2T_M\sqrt{ET\log(2T)} \tag{149}$$

$$\leq \sum_{e=1}^{E}\sum_{s,a} \nu_e(s,a)2\sqrt{\frac{14S\log(2AT)}{N_e(s,a)}}\|h_\pi^{\tilde{P},r}(\cdot)\|_\infty + 2T_M\sqrt{ET\log(2T)} \tag{150}$$

$$\leq 2T_M\sqrt{14S\log(2AT)}\sum_{s,a}\sum_{e=1}^{E}\frac{\nu_e(s,a)}{\sqrt{N_e(s,a)}} + 2T_M\sqrt{ET\log(2T)} \tag{151}$$

$$\leq 2(\sqrt{2}+1)T_M\sqrt{14S\log(2AT)}\sum_{s,a}\sqrt{N(s,a)} + 2T_M\sqrt{ET\log(2T)} \tag{152}$$

$$\leq 2(\sqrt{2}+1)T_M\sqrt{14S\log(2AT)}\sqrt{SAT} + 2T_M\sqrt{ET\log(2T)} \tag{153}$$

where Equation (113) follows from Lemma 4. Equation (114) follows from Lemma 2 with probability $1 - 1/T^5$. Equation (115) comes from Lemma 5. Equation (116) follows from [Jaksch et al., 2010, Lemma 19] and Equation (117) follows from Cauchy-Schwarz inequality.

Together with Equation (108), we get:

$$R_1(T) \leq 2(\sqrt{2}+1)T_M S\sqrt{AT\log(AT)} + 2T_M\sqrt{ET\log(2T)} + \frac{2T_M SE}{1-\rho} + E + \sqrt{T} \tag{154}$$

Combining $R_1(T)$ and $R_2(T)$ we get the required bound on regret. The bound on constraint violations follows similarly. □

*Proof of Lemma 1.* We begin with considering the policy $\pi$ in Assumption 3. We now prove the result for one $k \in [K]$ and the result follows for all $k \in [K]$. We consider an MDP with transition dynamics $P_e^k$ which maximizes $\zeta_\pi^{P',k}$ for all $\|P'(\cdot|s,a)-P(\cdot|s,a)\|_1 \leq \sqrt{\frac{14S\log(2At)}{N_e(s,a)}}$ for all $s,a$. Consider the difference between the average cost $k$ incurred from following policy $\pi$ on the MDP with true transition probabilities $P$ and the average cost $k$ incurred from following policy $\pi$ on the MDP with transition probabilities $P_e^k$ and using Lemma 3. We have:

$$\zeta_\pi^{\tilde{P}_e,k} - \zeta_\pi^{P,k} \leq \zeta_\pi^{P_e^k,k} - \zeta_\pi^{P,k} \tag{155}$$

$$= \sum_{s,a} d_\pi^P(s,a)B_\pi^{P_e^k,k}(s,a) \tag{156}$$

$$= \mathbb{E}\left[B_\pi^{P_e^k,k}(s,a)\right] \tag{157}$$

where the of Bellman error $B_\pi^{\tilde{P}_e^k,k}(s,a)$ is of the following form,

$$B_\pi^{\tilde{P}_e,k}(s,a) = \lim_{\gamma\to 1}\left(Q_{\gamma,\pi}^{\tilde{P},k}(s,a) - c^k(s,a) - \gamma\sum_{s'\in\mathcal{S}}P(s'|s,a)V_{\gamma,\pi}^{\tilde{P},k}(s,a)\right),$$

and the value function, $V_{\gamma,\pi}^{\tilde{P},k}(s)$ and $Q$-value, $Q_{\gamma,\pi}^{\tilde{P},k}(s,a)$, function become:

$$V_{\gamma,\pi}^{\tilde{P},k}(s) = \sum_{t=1}^{\infty}\gamma^{t-1}\mathbb{E}_{a_t\sim\pi,s_{t+1}\sim P}\left[c^k(s_t,a_t)|s_1=s\right]$$

$$Q_{\gamma,\pi}^{\tilde{P},k}(s,a) = \sum_{t=1}^{\infty}\gamma^{t-1}\mathbb{E}_{a_t\sim\pi,s_{t+1}\sim P}\left[c^k(s_t,a_t)|s_1=s,a_1=a\right].$$

We bound the expectation using Azuma-Hoeffdings inequality as follows:

$$\mathbb{E}\left[B_\pi^{\tilde{P}_e^k,k}(s,a)\right] = \mathbb{E}\left[B_\pi^{\tilde{P}_e^k,k}(s_t,a_t)|\mathcal{H}_{t_e-1}\right] + C\|h_\pi^{P_e,k}(\cdot)\|_\infty\rho^{t-t_e} \tag{158}$$

$$= \frac{1}{t_{e+1}-t_e}\sum_{t=t_e}^{t_{e+1}-1}\left(\mathbb{E}\left[B_\pi^{\tilde{P}_e^k,k}(s_t,a_t)|\mathcal{H}_{t_e-1}\right] + C\|h_\pi^{P_e,k}(\cdot)\|_\infty\rho^{t-t_e}\right) \tag{159}$$

$$\leq \frac{1}{t_{e+1}-t_e}\sum_{t=t_e}^{t_{e+1}-1}\left(\mathbb{E}\left[B_\pi^{\tilde{P}_e^k,k}(s_t,a_t)|\mathcal{H}_{t_e-1}\right]\right) + \frac{CS\|h_\pi^{P_e,k}(\cdot)\|_\infty}{(1-\rho)(t_{e+1}-t_e)} \tag{160}$$

$$\leq \frac{1}{t_{e+1}-t_e}\left(T_M\sqrt{14S\log AT}\sum_{s,a}\frac{\nu_e(s,a)}{\sqrt{N_e(s,a)}} + 4T_M\sqrt{7(t_{e+1}-t_e)\log(t_{e+1}-t_e)}\right)$$
$$+ \frac{CST_M}{(1-\rho)(t_{e+1}-t_e)} \tag{161}$$

$$\leq \frac{1}{t_{e+1}-t_e}\left(T_M\sqrt{14S\log AT}\sum_{s,a}\sqrt{\nu_e(s,a)} + 4T_M\sqrt{7(t_{e+1}-t_e)\log(t_{e+1}-t_e)}\right)$$
$$+ \frac{CST_M}{(1-\rho)(t_{e+1}-t_e)} \tag{162}$$

$$\leq \frac{1}{t_{e+1}-t_e}\left(T_MS\sqrt{14A\log AT}\sqrt{\sum_{s,a}\nu_e(s,a)} + 4T_M\sqrt{7(t_{e+1}-t_e)\log(t_{e+1}-t_e)}\right)$$
$$+ \frac{CST_M}{(1-\rho)(t_{e+1}-t_e)} \tag{163}$$

$$\leq \frac{1}{t_{e+1}-t_e}\left(T_MS\sqrt{14A\log AT}\sqrt{(t_{e+1}-t_e)} + 4T_M\sqrt{7(t_{e+1}-t_e)\log(t_{e+1}-t_e)}\right)$$
$$+ \frac{CST_M}{(1-\rho)(t_{e+1}-t_e)} \tag{164}$$

$$\leq \left(T_MS\sqrt{\frac{14A\log AT}{(t_{e+1}-t_e)}} + 4T_M\sqrt{\frac{7\log(t_{e+1}-t_e)}{(t_{e+1}-t_e)}}\right) + \frac{CST_M}{(1-\rho)(t_{e+1}-t_e)} \tag{165}$$

where Equation (123) is obtained by summing both sides from $t=t_e$ to $t=t_{e+1}$. Equation (124) is obtained by summing over the geometric series with ratio $\rho$. Equation (125) comes from analysis used in the proof of Theorem 1. Equation (126) comes from the fact that $N_e(s,a) \geq \nu_e(s,a)$ for all $s,a$, and then replacing the lower bound of $N_e(s,a)$. Equation (127) follows from the Cauchy Schwarz inequality. Equation (128) follows from the fact that the epoch length $t_{e+1}-t_e$ is same as the number of visitations to all state action pairs in an epoch.

Combining Equation (129) with Equation (121), we obtain the required result as follows:

$$\zeta_\pi^{\tilde{P}_e, k} \leq \zeta_\pi^{P_e^k, k} - \zeta_\pi^{P, k} + \zeta_\pi^{P, k} \tag{166}$$

$$\leq \left( T_M S \sqrt{\frac{14 A \log AT}{(t_{e+1} - t_e)}} + 4 T_M \sqrt{\frac{7 \log(t_{e+1} - t_e)}{(t_{e+1} - t_e)}} \right) + \frac{CST_M}{(1 - \rho)(t_{e+1} - t_e)} + \zeta_\pi^{P, k} \tag{167}$$

$$\leq \left( T_M S \sqrt{\frac{14 A \log AT}{\sqrt{T}}} + 4 T_M \sqrt{\frac{7 \log(\sqrt{T})}{\sqrt{T}}} \right) + \frac{CST_M}{(1 - \rho)\sqrt{T}} + \zeta_\pi^{P, k} \tag{168}$$

$$\leq \left( T_M S \sqrt{\frac{14 A \log AT}{\sqrt{T}}} + 4 T_M \sqrt{\frac{7 \log(\sqrt{T})}{\sqrt{T}}} \right) + \frac{CST_M}{(1 - \rho)\sqrt{T}} + C_k - \kappa \tag{169}$$

$$\leq C_k \tag{170}$$

where Equation (132) comes from the fact that we consider epoch length $t_{e+1} - t_e \geq \sqrt{T}$ and Equation (133) comes from Assumption 3 and Equation (134) comes from the value of Slater's constant $\kappa$ in Assumption 3. Replicating the analysis for all $k \in [K]$, for the policy $\pi$, $\zeta_\pi^{\tilde{P}_e, k}$ satisfy the constraint for all $k \in [K]$ and hence, the optimization problem in Equation (14)-(17) is feasible. □