# OpenReview forum: "Regret Guarantees for Model-Based Reinforcement Learning with Long-Term Average Constraints"
_auai.org/UAI/2022/Conference — UAI 2022 Poster_

### Official Review · Reviewer_SGRP · 2022-03-24

**Q2(1) Originality/Novelty:** 2
**Q2(2) Significance/Impact:** 2
**Q2(3) Correctness/Technical Quality:** 4
**Q2(6) Clarity Of Writing:** 3
**Q6 Overall Score:** 5
**Q8 Confidence In Your Score:** 3

**Q1 Summary And Contributions:**

This paper studied regret bounds for learning in average-reward ergodic MDP with long-term average constraints. Sublinear regret is given under TS policy.

**Q2 Assessment Of The Paper:**

More detailed information regarding each of these aspects is given below:

**Q2(5) Reproducibility:**

2: Fair: Key resources (e.g., proofs, code, data) are unavailable but key details (e.g., proof sketches, experimental setup) are sufficiently well-described for an expert to confidently reproduce the main results.

**Q3 Main Strengths:**

This paper extends the regret guarantee of constrained MDP problem from episodic case to infinite-horizon undiscounted case. This is the first kind of analysis and should be important.




**Q4 Main Weakness:**

In my view, the biggest concern of this work is the unichain assumption and the tightness of the bound. Constrained MDP is not new and regret guarantee is not new for episodic MDP. And this paper focused only on tabular cases so the assumption and tightness is very important. For standard regret minimization for average-reward problems, either UCRL2 or TS can already achieve sublinear regret bound under weakly-communicating MDP which is much weaker than unichain MDP. That means this work has a weaker result if we remove the constraint. This limits the contribution of this work.

This tightness of the bound is not studied as well. Even for unichain or ergodic MDP, you still bound the regret by diameter which seems to be loose. Since the MDP can explore by itself, I am wondering do we actually need TS?


**Q5 Detailed Comments To The Authors:**

See above.

**Q7 Justification For Your Score:**

The theoretical contribution for this paper might be limited due to the tightness of the bound and the assumption.


**Q9 Complying With Reviewing Instructions:**

1: Yes.

---

### Official Review · Reviewer_qD3K · 2022-04-10

**Q2(1) Originality/Novelty:** 2
**Q2(2) Significance/Impact:** 3
**Q2(3) Correctness/Technical Quality:** 3
**Q2(6) Clarity Of Writing:** 2
**Q6 Overall Score:** 6
**Q8 Confidence In Your Score:** 3

**Q1 Summary And Contributions:**

This paper studies the MDP with long-term averaged cost-constrained, where the agent aims to maximize the infinite-time averaged cumulative reward and satisfies the constraints on the cost. The problem can be reduced to linear programming when the transition function is known. The main contribution of this paper is to use the Thompson sampling for estimating the transition matrix and shows an $tilde{O}\sqrt{T}$ regret bound on the reward and the costs.

**Q2 Assessment Of The Paper:**

More detailed information regarding each of these aspects is given below:

**Q2(4) Quality Of Experiments (Optional):**

2: Fair: The experimental evaluation is weak: important baselines are missing, or the results do not adequately support the main claims.

**Q2(5) Reproducibility:**

3: Good: Key resources (e.g., proofs, code, data) are available and key details (e.g., proofs, experimental setup) are sufficiently well-described for competent researchers to confidently reproduce the main results.

**Q3 Main Strengths:**

The main strength of this paper is to provide an $O(\sqrt{T})$ regret bound for the MDP with long-term averaged constraint. Under the same problem setup, the $O(\sqrt{T})$ regret bound on both reward and the costs significantly improves the previous results $O(T^{2/3})$ in [Singh et al., 2020].


**Q4 Main Weakness:**

The weakness of this paper is as follows:

- about the clarity and novelty: when the transition is known, [Singh et al., 2020] has already shown that the CMDP problem can be solved with LP. The main difference between the proposed method (CMDP-PSRL) with that of [Singh et al., 2020] (CDMP-UCRL) is that CMDP-PRSL estimates the transition function with Thompson sampling while CMDP-PRSL use UCRL. It is unclear and surprising to me how this change can lead to such a significant improvement in the regret bound, since Thompson sampling and OFU-based algorithm usually enjoy regret bound with the same dependence on T.

- about the experiments: there is no other method compared in the experimental part. To show the superiority of the proposed method, I think it is necessary to compare the proposed method with the closely related method CMDP-UCRL [Singh et al., 2020].


**Q5 Detailed Comments To The Authors:**

This paper has achieved O(\sqrt{T}) regret bounds for the CMDP problem, which significantly improves the O(T^{2/3}) bound in [Singh et al., 2020]. However, I am still concerned about the clarity of this paper.

As I have mentioned in Q4, it seems to me that the main difference between this paper and [Singh et al., 2020] is that the proposed method uses Thompson sampling (TS) to estimate the transition instead of UCRL [Singh et al., 2020]. It is unclear why the change of the estimator could lead to such a significant improvement. I am still checking the proof. I think it would be better for the author to highlight the key insight behind the improvement.


Meanwhile, I find the O(\sqrt{T^{1.5}}) regret bound for [Singh et al., 2020] cited in this paper is different from the arXiv version (https://arxiv.org/pdf/2002.12435.pdf), where the regret bound is O(T^{2/3}). I think it would be better for the author to make this issue clear.

--minor point--
- last second line in the abstract: a O (T) regret bounds --> O (T) regret bounds
- 9th line of the first paragraph: has --> have
- 11th line of the second paragraph: say --> saying
- last paragraph in page 2: O(T^{1.5}) --> O(T^{2/3}) or O(\sqrt{T^{1.5}})?


**Q7 Justification For Your Score:**

The O(\sqrt{T}) regret bound improves the O(T^{2/3}) result in the previous work. However, it is still unclear how a change in the transition estimator from the UCRL to Thompson sampling could lead to such a significant improvement. I will raise my score if the author makes this clear.

**Q9 Complying With Reviewing Instructions:**

1: Yes.

---

### Official Review · Reviewer_VYQn · 2022-04-14

**Q2(1) Originality/Novelty:** 1
**Q2(2) Significance/Impact:** 2
**Q2(3) Correctness/Technical Quality:** 3
**Q2(6) Clarity Of Writing:** 3
**Q6 Overall Score:** 5
**Q8 Confidence In Your Score:** 3

**Q1 Summary And Contributions:**

This paper studies model-based RL algorithms for infinite-horizon CMDP. The proposed algorithm maintains a posterior distribution of the unknown transition probability and finds the optimal constrained policy of the sampled transition dynamic. The authors show that their algorithm enjoys a sublinear regret in both reward and constraint.

**Q2 Assessment Of The Paper:**

More detailed information regarding each of these aspects is given below:

**Q2(4) Quality Of Experiments (Optional):**

3: Good: The experimental evaluation is adequate, and the results convincingly support the main claims.

**Q2(5) Reproducibility:**

3: Good: Key resources (e.g., proofs, code, data) are available and key details (e.g., proofs, experimental setup) are sufficiently well-described for competent researchers to confidently reproduce the main results.

**Q3 Main Strengths:**

+ The presentation is clear.
+ The technical approach is reasonable.
+ The experiment results are clear.


**Q4 Main Weakness:**

- The technical challenges and novelties are unknown.

**Q5 Detailed Comments To The Authors:**

Compared with RL for episodic CMDP, there are two differences and challenges for the setting considered in this work: how to build bounds for estimated posterior sampling MDP and true MDP, and how to deal with the long-term average regret. The authors may want to highlight the key differences between their work and previous works for CMDP, like Qiu et al. (2021) or Kalagarla et al. (2020). Specifically, what is the challenge of directly applying UC-CFH in Kalagarla et al. (2020) to the infinite-horizon CMDP setting?

- Above (4), is find -> finding


**Q7 Justification For Your Score:**

This work proposed a posterior sampling-based RL algorithm for infinite-horizon CMDP and obtained sublinear regret for both reward and constraints, which is the first result in the specific problem setting. However, given existing discussions, I can not tell if there exist any unique challenges to simply modifying an existing algorithm from episodic setting to infinite-horizon setting. Thus, it is hard to judge the theoretical contribution of this paper and I recommend a boarderline accept.

**Q9 Complying With Reviewing Instructions:**

1: Yes.

---

### Decision · Program_Chairs · 2022-05-15

**Decision:**

Accept (Poster)

**Comment:**

Meta Review: This paper makes a contribution to the understanding of exploration in constrained MDPs. It proposes a new algorithm to explore a constrained MDP using posterior sampling and offers a tighter regret-bound than exists in the literature. The reviewers generally felt that this paper made a reasonable contribution, but expressed some concerns about the assumptions and exposition in the paper. The authors did address some of these questions in their rebuttal. Specifically, the paper does not do a very good job at explaining the intuition behind the improved regret bound, which relies on the inherent exploratory behavior in a unichain MDP. The authors do explain this in the paper, but should make clear how this manifests in the improved bound. The second concern is that the unichain assumption is very strong—many (most) practical problems are not unichain (though clearly some are). The authors simply assert this assumption and offer no justification. The unichain assumption certainly limits the applicability of the result, but moreover, the settings in which the assumption might make sense are not articulated. Finally, the empirical results are quite weak—no baseline approaches are considered.